# The Importance of Online Data:
# Understanding Preference Fine-tuning via Coverage

**Yuda Song**
Carnegie Mellon University
yudas@cs.cmu.edu

**Gokul Swamy**
Carnegie Mellon University
gswamy@cs.cmu.edu

**Aarti Singh**
Carnegie Mellon University
aarti@cs.cmu.edu

**J. Andrew Bagnell**
Aurora Innovation, Carnegie Mellon University
dbagnell@aurora.tech

**Wen Sun**
Cornell University
ws455@cornell.edu

## Abstract

Learning from human preference data has emerged as the dominant paradigm for fine-tuning large language models (LLMs). The two most common families of techniques – online reinforcement learning (RL) such as Proximal Policy Optimization (PPO) and offline contrastive methods such as Direct Preference Optimization (DPO) – were positioned as equivalent in prior work due to the fact that both have to start from the same offline preference dataset. To further expand our theoretical understanding of the similarities and differences between online and offline techniques for preference fine-tuning, we conduct a rigorous analysis through the lens of *dataset coverage*, a concept that captures how the training data covers the test distribution and is widely used in RL. We prove that a global coverage condition is both necessary and sufficient for offline contrastive methods to converge to the optimal policy, but a weaker partial coverage condition suffices for online RL methods. This separation provides one explanation of why online RL methods can perform better than offline methods, especially when the offline preference data is not diverse enough. Finally, motivated by our preceding theoretical observations, we derive a hybrid preference optimization (HyPO) algorithm that uses offline data for contrastive-based preference optimization and online unlabeled data for KL regularization. Theoretically and empirically, we demonstrate that HyPO is more performant than its pure offline counterpart DPO, while still preserving its computation and memory efficiency.

## 1 Introduction

Due to the difficulty of manually specifying reward functions for complex tasks [7], preference-based learning has emerged as a critical component in the fine-tuning procedure for large language models (LLMs) [40, 30, 45, 44]. There are two predominant flavors of preference learning for LLMs: online reinforcement learning (RL) methods such as PPO [11, 30] and offline contrastive methods like Direct Preference Optimization (DPO) [33] and Identity Preference Optimization (IPO) [3].

Online RL methods usually follow the two-stage procedure prescribed in [30]: one first trains a reward model (classifier) on a fixed offline preference dataset before using it to provide reward labels for on-policy generations, which are then fed to a downstream RL algorithm like Proximal Policy Optimization (PPO) [36]. Since the reward model is learned from static offline preference data, to avoid over-optimizing the reward model [17], one typically adds a reverse KL penalty to encourage the model to stay close to some reference policy. We will refer to this procedure as reinforcement learning from human feedback (RLHF) in this paper. While empirically performant, RLHF requires repeated querying of the reward model (which is often itself an LLM) as well as sampling from the current policy. In response to the computational expense and relatively complex nature of this

38th Conference on Neural Information Processing Systems (NeurIPS 2024).

procedure, purely offline methods like DPO [33] and IPO [3] have been proposed as alternative methods for preference fine-tuning. These methods do not need to fit separate reward models, instead opting to simply train the policy directly on the offline preference dataset via a ranking loss.

Offline contrastive methods like DPO are usually derived via applying a reparameterization trick to the closed-form solution of the minimum relative entropy problem [63] that RLHF techniques attempt to approximate. Thus, several authors have described these methods as equivalent (at least in theory) to the standard RLHF procedure [33, 3]. However, recent (mostly empirical) work has contradicted this perspective: [43] find that online methods out-perform offline methods and attribute this fundamentally to on-policy sampling, [55] argues that the online RL methods produce an often desirable subset of the possible DPO loss minimizers, and [42] provide empirical support for the claim that online and contrastive training provide orthogonal benefits. However, a rigorous theoretical separation is still lacking in the pre-existing literature, which motivates our key questions:

> *What is the statistical separation between the online RLHF method and offline*
> *contrastive methods? What causes this separation and what does it imply?*

To answer these questions, we focus on the coverage of the preference dataset, a key concept that is widely used in RL [22, 28, 58] for analyzing the impact of offline or exploratory data distributions. Through the lens of coverage of the offline preference dataset, we make the following contributions:

- We prove that the global coverage condition [28], the strongest possible coverage condition in RL, is necessary for offline contrastive algorithms like DPO to converge to the optimal policy. In contrast, we identify a weaker local coverage condition that is sufficient for online RLHF algorithms, thus provably separating the two types of algorithms. The separation is due to the difference in reward modeling and on/offline regularization – in short, *there is no free lunch from bypassing explicit reward learning and online rollouts*. As global coverage might sometimes be violated in practice, our separation result can perhaps explain why RLHF works better than offline methods [42, 43, 56].
- Although offline contrastive methods are derived from a reverse-KL objective, we prove that the policies trained via offline methods can still have infinite reverse-KL in the partial coverage setting. In contrast, we show that RLHF can always control the reverse KL via directly optimizing reverse KL using online samples. This means that on realistic problems, RLHF has stronger guarantees for remaining close to the reference policy than offline contrastive methods.
- We propose Hybrid Preference Optimization (HyPO) to address the deficiencies of offline contrastive methods while maintaining some of their computational simplicity. HyPO is a hybrid RL algorithm [51, 39] where offline data is used for the DPO objective while online samples are used to explicitly control the reverse KL divergence to the reference policy. We empirically demonstrate that HyPO outperforms DPO, on the *TL;DR* summarization task [40] on all metrics including both the GPT4 win-rate and the reverse KL divergence to the reference policy, and on general chat benchmarks such as AlpacaEval 2.0 [15], trained with the UltraFeedback dataset [14]. In addition, HyPO also mitigates the overfitting issues observed in the offline constrastive based methods [43].
- We provide an explanation of why RLHF and offline contrastive methods decrease the probability of both preferred and rejected responses during training. In particular, under our function approximation-based global coverage condition, we show that such behavior is actually desirable for DPO and RLHF policies to extrapolate and generalize to optimal actions that do not appear in the training dataset. However, without function approximation, algorithms like DPO can mistakenly increase the likelihood of sub-optimal actions. This establishes the importance of function approximation for the success of the algorithms such as DPO.

Taken together, our results establish the critical role *coverage* plays in terms of convergence properties of preference learning algorithms as well as in the design of new, performant empirical approaches.

## 2 Related Work

**Preference Fine-Tuning (PFT).** As discussed in the introduction of our work, there are two major paradigms for preference fine-tuning of LLMs. The first one, online RL methods [30], proposes to first train a reward model (classifier) to predict human preferences, followed by running an RL method to optimize this learned reward function. While PPO [36] is the most popular RL algorithm used in the online RLHF framework by far [40, 30, 45], more recent work by [1] shows that simpler online RL algorithms like REINFORCE [48] also work well. The second class of methods, offline contrastive techniques [60, 33, 3], avoid explicit reward modeling and directly optimize their objective on the offline preference dataset. Recently there are *hybrid* methods that combine offline

preference data with online preference labels [52, 19, 35, 3] – we leave extending our analysis to this setting to future work. Throughout our paper, we assume for simplicity of analysis that preferences are generated by an underlying utility function and therefore contain no intransitivities [29, 41].

**Understanding PFT.** Prior work has studied different parts of the standard RLHF recipe [17, 23, 38, 16] and the impact of preference data quality [37]. In our work, we instead take a converge-based perspective on the relationship between online RL methods and offline contrastive methods. Although derived from the same minimum relative entropy objective [63] and perceived as equivalent by some early work [33, 3], more recent work has started to unravel the distinctions between these two classes of methods. [43] repeatedly observe better performance from online rather than offline methods and after rigorously validating a variety of hypotheses, conclude that on-policy sampling is indispensable for ensuring a high quality policy. [42] perform an in-depth study of the effects of preference data, contrastive losses, and on-policy sampling and conclude that a combination of contrastive losses and interactive training is most preferable in practice. [55] also observe better performance from online PPO than from offline DPO and argue this is because the former is able to eliminate a larger set of policies that are undesirable from the perspective of the later. We supplement these mostly empirical observations with a rigorous theoretical explanation for the observed behavior through the lens of dataset coverage, as well as designing an algorithm that addresses the key weaknesses of offline contrastive approaches.

We defer additional related works to Appendix A.

## 3 Preliminaries

Following a wide range of recent works [33, 3], we consider the RLHF problem in the contextual bandit formulation [25]. This is a reasonable simplification, as one can consider the generated sequence of tokens as one single action, due to the fact that the states are the generated tokens, and the dynamics are deterministic. We denote the context (prompt) space as $\mathcal{X}$, and the action (response) space as $\mathcal{Y}$. Note that due to the finiteness of the possible tokens, the action space is finite but combinatorially large. We use $\rho \in \Delta(\mathcal{X})$ to denote the distribution of the prompts, and $\pi : \mathcal{X} \to \Delta(\mathcal{Y})$ as policies (LLMs) that map prompts to a distribution of responses. We also consider the reward function class $\mathcal{R} : \mathcal{X} \times \mathcal{Y} \to \mathbb{R}$, which assigns a reward to each context-response pair.

We assume access to a reference policy $\pi_{\mathsf{ref}}$, which is usually referred to as the policy learned using supervised data when training the LLM, that needs to be further fine-tuned to align with human values. An offline preference dataset is collected in the format of $\mathcal{D} = \{x, y^+, y^-\}$ triplets: given context $x \sim \rho$, the preference policy samples two responses $y^1, y^2 \sim \mu(\cdot \mid x)$, where $\mu$ is the offline response distribution. Previous works assume either $\mu$ to be the same distribution as $\pi_{\mathsf{ref}}$ [33] or different offline distribution [3, 35, 18]. Then, $y^1$ is labelled as $y^+$ (thus $y^2$ as $y^-$) with probability $p^*(y^1 \succ y^2 \mid x)$, where $p^*$ is defined by the Bradley-Terry model [6]:

$$p^*(y^1 \succ y^2 \mid x) = \frac{\exp(r^*(x, y^1))}{\exp(r^*(x, y^1)) + \exp(r^*(x, y^2))},$$

where $r^*$ is the human's implicit reward function. Note that this rules out intransitive preferences [41, 29]. Throughout the paper, we will make the following assumption on the reward function:

**Assumption 3.1** (Boundedness of the reward). $\|r^*\|_\infty \leq R$.

In many previous works, this formulation has been the canonical way to model the preference data in the RLHF literature [11, 33, 3]. The goal is to learn a policy $\pi$ to maximize the objective $J(\pi)$, where

$$J(\pi) = \mathbb{E}_{x \sim \rho}\left[\mathbb{E}_{y \sim \pi(\cdot \mid x)}[r^*(x, y)] - \beta \mathsf{KL}(\pi(\cdot \mid x) \| \pi_{\mathsf{ref}}(\cdot \mid x))\right], \tag{1}$$

i.e., we want to both maximize the human implicit reward, and not deviate too much from the reference policy. We denote the optimal policy $\pi^* \in \operatorname{argmax}_{\pi \in \Pi} J(\pi)$. Here we call $\mathsf{KL}(\pi(\cdot \mid x) \| \pi_{\mathsf{ref}}(\cdot \mid x))$ reverse KL because $\pi$ – the policy to be optimized, appears first. We will call $\mathsf{KL}(\pi_{\mathsf{ref}}(\cdot \mid x) \| \pi(\cdot \mid x))$ forward KL. By the definition of KL, we have

Definition of reverse KL: $\quad \mathsf{KL}(\pi(\cdot \mid x) \| \pi_{\mathsf{ref}}(\cdot \mid x)) := \mathbb{E}_{y \sim \pi(\cdot \mid x)}[\ln(\pi(y|x)/\pi_{\mathsf{ref}}(y|x))]. \tag{2}$

Note that the expectation in reverse KL is under $\pi$, indicating that evaluating and optimizing reverse KL requires drawing *online samples* from $\pi$. In contrast, evaluating forward KL only requires *offline*

*samples* drawn from $\pi_{\mathsf{ref}}$. As we will show, this key difference between reverse KL and forward KL plays an important role of separating online RLHF and offline contrastive methods such as DPO. In this paper, we consider two types of algorithms: online RL-based algorithms, and offline contrastive-based algorithms.

**Online RLHF Algorithms.** We consider algorithms such as [11, 1] as the online RL based methods. We abstract these algorithms as the following procedure: the algorithm performs the following two-stage procedure: one first trains a reward model $\widehat{r}$ that minimizes the Bradley-Terry loss [1]

$$\widehat{r} \in \underset{r \in \mathcal{R}}{\operatorname{argmax}} \widehat{\mathbb{E}}_{x,y^+,y^- \sim \mathcal{D}} \left[ \log \left( \frac{\exp(r(x,y^+))}{\exp(r(x,y^+)) + \exp(r(x,y^-))} \right) \right], \tag{3}$$

and perform policy optimization (such as PPO [36]) to optimize the policy with the reward model $\widehat{r}$:

$$\pi_{\mathsf{rlhf}} \in \underset{\pi}{\operatorname{argmax}} \widehat{\mathbb{E}}_{x \sim \mathcal{D}} \left[ \mathbb{E}_{y \sim \pi(\cdot|x)}[\widehat{r}(x,y)] - \beta \mathsf{KL}(\pi(\cdot \mid x) || \pi_{\mathsf{ref}}(\cdot \mid x)) \right].$$

However, this policy optimization step requires extensive online sampling, and possibly training an additional critic model (e.g., PPO), in addition to the reward model and policy.

**Offline Contrastive Algorithms.** To circumvent the above-mentioned computational burden, several purely offline contrastive-based methods (i.e., without RL) have been proposed. In this paper, we focus on the following two most representative methods. The first is Direct Preference Optimization (DPO) [33], where the objective is $\pi_{\mathsf{dpo}} \in \operatorname{argmax}_\pi \ell_{\mathsf{dpo}}(\pi)$ with

$$\ell_{\mathsf{dpo}}(\pi) = \widehat{\mathbb{E}}_{x,y^+,y^- \sim \mathcal{D}} \left[ \log \left( \frac{\exp\left(\beta \log\left(\frac{\pi(y^+|x)}{\pi_{\mathsf{ref}}(y^+|x)}\right)\right)}{\exp\left(\beta \log\left(\frac{\pi(y^+|x)}{\pi_{\mathsf{ref}}(y^+|x)}\right)\right) + \exp\left(\beta \log\left(\frac{\pi(y^-|x)}{\pi_{\mathsf{ref}}(y^-|x)}\right)\right)} \right) \right]. \tag{4}$$

Another offline contrastive method we will discuss in our paper is Identity Preference Optimization [3], but we will defer its technical details to the appendix.

## 4 Offline Contrastive Methods Require a Stronger Coverage Condition than Online RL Methods

We start by introducing the mathematical formulation of the coverage framework. The strongest coverage condition is the following global coverage condition [28]: we say any offline distribution $\mu$ covers a policy $\pi$ if we have $\max_{x,y:\rho(x)>0} \frac{\pi(y|x)}{\mu(y|x)} \leq C_{\mathsf{glo}}$. Throughout this section, we will adopt the setting where $\mu = \pi_{\mathsf{ref}}$ [33]. Formally, we assume the following condition:

**Assumption 4.1** (Global Coverage). *For all $\pi$, we have*

$$\max_{x,y:\rho(x)>0} \frac{\pi(y \mid x)}{\pi_{\mathsf{ref}}(y \mid x)} \leq C_{\mathsf{glo}}.$$

For the coverage terms, we always adopt the convention that $\frac{0}{0} = 0$. Note that one sufficient condition for this assumption is that, for any prompt $x$, and any token sequence $y$, we have $\pi_{\mathsf{ref}}(y \mid x) \geq 1/C_{\mathsf{glo}}$.

As has been recognized in the offline RL literature, global coverage is a strong assumption, and efforts have been made to circumvent this assumption with more relaxed coverage conditions [46, 10, 58]. In this paper, we will consider the following partial coverage assumption that is weaker than Assumption 4.1:

**Assumption 4.2** (Local KL-ball Coverage). *For all $\varepsilon_{\mathsf{kl}} < \infty$ and all policy $\pi$ such that* $\mathbb{E}_{x \sim \rho}[\mathsf{KL}(\pi(\cdot \mid x) || \pi_{\mathsf{ref}}(\cdot \mid x))] \leq \varepsilon_{\mathsf{kl}}$, *we have*

$$\max_{x,y:\rho(x)>0} \frac{\pi(y \mid x)}{\pi_{\mathsf{ref}}(y \mid x)} \leq C_{\varepsilon_{\mathsf{kl}}}.$$

Note that $C_{\varepsilon_{\mathsf{kl}}}$ depends on $\varepsilon_{\mathsf{kl}}$. This coverage notion is relatively new in the RL literature and only appeared in previous analysis of RLHF algorithms [9]. We call this local coverage condition since it only requires $\pi_{\mathsf{ref}}$ to cover the policies that is within some KL-divergence ball centered at $\pi_{\mathsf{ref}}$. The intuition of this assumption is, for any algorithm that can control the reverse KL of the output policy, we can leverage the coverage condition to relate the error under the output policy to its error under the offline distribution, and thus guarantee its performance. Finally, we note that since the policies with bounded KL is a subset of all policies, for a fixed $\pi_{\mathsf{ref}}$, we always have $C_{\varepsilon_{\mathsf{kl}}} \leq C_{\mathsf{glo}}$.

---

[1]We use $\widehat{\mathbb{E}}$ to denote the empirical expectation over the dataset.

**Remark 4.1.** Taking a closer look at Assumption 4.2, we can see that this assumption is always true in the sense that for any policy with $\varepsilon_{\mathsf{kl}} < \infty$, $\max_{x,y:\rho(x)>0} \frac{\pi(y|x)}{\pi_{\mathsf{ref}}(y|x)} < \infty$, i.e., $C_{\varepsilon_{\mathsf{kl}}} < \infty$, for any $\varepsilon_{\mathsf{kl}}$. However, while being bounded, $C_{\varepsilon_{\mathsf{kl}}}$ can be large. Indeed a simple calculation can show that $\max_{x,y:\rho(x)>0} \frac{\pi(y|x)}{\pi_{\mathsf{ref}}(y|x)}$ can be as large as $\max_{x,y:\pi(y|x)>0} \exp\left(\frac{\varepsilon_{\mathsf{kl}}}{\pi(y|x)}\right)$. This can be undesirable because this suggests bounded reverse KL itself is not enough to guarantee optimality: the error can have an *exponential* amplification when switching from $\pi_{\mathsf{ref}}$ to $\pi$. Thus this motivates Assumption 4.2, which assumes that $C_{\varepsilon_{\mathsf{kl}}}$ is reasonably small.

In what follows, we will show that the global coverage assumption (Assumption 4.1) is necessary for offline contrastive-based algorithms such as DPO and IPO, but partial coverage assumption such as Assumption 4.2 is sufficient for online RL based algorithms. This establishes a separation between the two types of algorithms. We emphasize this theoretical separation explains why in practice online methods is less prone to problems such as reward hacking and producing out-of-distribution responses that are due to dataset with insufficient coverage.

## 4.1 Global Coverage is Necessary for Offline Contrastive Algorithms

**Failure of DPO Under Partial Coverage.** Now we show that if the strong coverage Assumption 4.1 breaks, then DPO can not guarantee any performance with respect to the objective function Eq. (1). The intuition is based on a rather common observation of the DPO algorithm: the DPO policy $\pi_{\mathsf{dpo}}$ may generate out of distribution responses, while in contrast, RLHF does not generate responses outside of the support of $\pi_{\mathsf{ref}}$ due to online reverse-KL constraint. For example, [55] provides a construction where $\pi_{\mathsf{dpo}}$ chooses a response where RLHF policy assigns 0 mass, thus proving that RLHF policies are a subset of DPO policies.

However, such construction assumes that the reward learning procedure of DPO makes arbitrarily large errors. Also, previous constructions assume deterministic preference, which is only true if the underlying reward function is unbounded. This violates the natural assumption of Assumption 3.1. In the following, we relax these constraints and thus show that DPO fails to guarantee any performance in a rather strong sense. Concretely, DPO constructs the following implicit reward class with the policy class $\Pi$: $\mathcal{R}_{\mathsf{dpo}} = \left\{ \beta \log\left(\frac{\pi(y|x)}{\pi_{\mathsf{ref}}(y|x)Z(x)}\right) \mid \pi \in \Pi \right\}$, where $Z(x)$ is a partition function that maps context to a real number and is independent of $y$. Plugging this formulation into the BT loss (Eq. (3)) recovers exactly the DPO loss (Eq. (4)) as the partition functions are canceled. Now we can characterize the returned policy by DPO as exactly whose corresponding reward function is accurate *in distribution*:

**Assumption 4.3** (In Distribution Reward Learning). *We assume the DPO policy $\pi_{\mathsf{dpo}}$ satisfies that:*

$$\mathbb{E}_{x,y\sim\rho\circ\pi_{\mathsf{ref}}}\left[\left(\beta\log\left(\frac{\pi_{\mathsf{dpo}}(y\mid x)}{\pi_{\mathsf{ref}}(y\mid x)Z(x)}\right) - r^*(x,y)\right)^2\right] \leq \varepsilon_{\mathsf{dpo}}.$$

Note that this is a rather strong assumption for BT loss – by Lemma B.2, at best one can only hope: for any learned reward function $\widehat{r}$, for each context $x$, there exists a constant $c(x)$ such that

$$\mathbb{E}_{x,y\sim\rho\circ\pi_{\mathsf{ref}}}\left[\left(\widehat{r}(x,y) - r^*(x,y) - c(x)\right)^2\right] \leq \varepsilon, \tag{5}$$

i.e., the reward model predicts the human reward up to a gap that is independent of $y$. This is due to the fact that BT loss only requires the reward function to capture the relative difference, or in other word, any constant shift (with respect to context) in the reward will be canceled in the BT loss. However, for the rest of the section, we will make the stronger learning assumption that the gap $c(x) = 0$ (such as in the case of Assumption 4.3). Previous counterexamples analysis violates this assumption, but we will show that even under this strong assumption, DPO still can not guarantee any performance.

**Proposition 4.1.** *Denote $\pi_{\mathsf{ref}}$ as any reference policy such that Assumption 4.1 breaks. Let $\Pi_{dpo}$ be the set of DPO returned policies such that Assumption 4.3 holds. Then there exists policy $\pi \in \Pi_{dpo}$ such that $J(\pi) = -\infty$.*

**Proof sketch.** Without loss of generality, we consider a promptless setting, and assume that the response space is $\mathcal{Y} = \{y_1, y_2, y_3\}$. Again without loss of generality, we assume $\pi_{\mathsf{ref}}$ only covers $y_1$ and $y_2$, and thus Assumption 4.1 breaks. We assume partition function $Z = 1$ for all $\pi$ but we will be

rigorous in the formal proof. Then consider the following policy $\pi$ such that

$$\beta \log \left( \frac{\pi(y_1)}{\pi_{\text{ref}}(y_1)} \right) = r^*(y_1) - \sqrt{\varepsilon_{\text{dpo}}}, \quad \text{and} \quad \beta \log \left( \frac{\pi(y_2)}{\pi_{\text{ref}}(y_2)} \right) = r^*(y_2) - \sqrt{\varepsilon_{\text{dpo}}},$$

One can check $\pi$ satisfies Assumption 4.3. Now consider the optimal policy $\pi^*(y_i) = \pi_{\text{ref}}(y_i) \exp \left( \frac{1}{\beta} r^*(y_i) \right)$, for $i \in \{1, 2\}$, and $\pi^*(y_3) = 0$. Since $\pi^*(y_1) + \pi^*(y_2) = 1$, combining everything we get $\pi(y_3) > 0$, which implies $\text{KL}(\pi || \pi_{\text{ref}})$ is unbounded, thus we complete the proof. $\square$

One can first relate the above construction to the parital coverage assumption Assumption 4.2: since the policy $\pi$ considered in the proof has unbounded reverse KL with respect to $\pi_{\text{ref}}$, thus it is not in the KL-ball of $\varepsilon_{\text{kl}}$ around $\pi_{\text{ref}}$, which implies that Assumption 4.2 is not sufficient for DPO. Next we show that global coverage is necessary for the IPO algorithm.

**Failure of IPO Under Partial Coverage.** To show that the global coverage is necessary for IPO, we can even assume a stronger in-distribution learning guarantee, that is, the returned policy achieves the smallest error on its population loss in distribution.

**Proposition 4.2** (Informal). *Denote $\pi_{\text{ref}}$ as any reference policy such that Assumption 4.1 breaks. Let $\Pi_{ipo}$ be the set of IPO returned policies such that it is the minimizer of in-distribution error on its population loss. Then there exists policy $\pi \in \Pi_{ipo}$ such that $J(\pi) = -\infty$.*

We defer the detailed setup and formal version to Appendix D, but the construction for the above proofs share the same intuition: the reverse KL term in the objective function can be unbounded. For offline contrastive-based algorithms, the KL regularization is only enforced under the data distribution, and thus the algorithm can not guarantee bounded reverse KL if the reference policy does not cover the response space well. Although we only showed counterexamples for DPO and IPO, we conjecture that the same intuition holds for other offline contrastive-based algorithms. One natural question at this point would be: how about the forward KL? Not surprisingly, the forward KL for DPO (but we conjecture for other offline constructive-based methods as well) is vacuously large, and we formalize this result in Appendix C.2.

**Remark 4.2.** The folklore that DPO is equivalent to RLHF is often based on some assumption that is much stronger than Assumption 4.3: it requires that the learned policy has a point-wise accuracy guarantee $\beta \ln(\pi_{\text{dpo}}(y|x)/\pi_{\text{ref}}(y|x)) = r^*(x, y)$ for all $x, y$. Such a point-wise guarantee is unrealistic in reality and does not hold in general in the supervised learning sense. The in-distribution style guarantee in Assumption 4.3 is the best one could hope for from a supervised learning algorithm.

## 4.2 Global Coverage is Sufficient for Offline Contrastive Algorithms

After showing that global coverage is necessary for DPO to guarantee any performance, we now show that it is sufficient for the performance guarantee.

**Theorem 4.1.** *Let $\pi_{\text{ref}}$ be any reference policy such that Assumption 4.1 holds. For any policy $\pi_{\text{dpo}}$ such that the event in Assumption 4.3 holds, we have that*

$$J(\pi^*) - J(\pi_{\text{dpo}}) = O(C_{\text{glo}} \sqrt{\varepsilon_{\text{dpo}}}).$$

**Proof.** By Lemma B.1, we have

$$J(\pi^*) - J(\pi_{\text{dpo}}) \leq \mathbb{E}_{x \sim \rho} \mathbb{E}_{y^1 \sim \pi^*(\cdot|x), y^2 \sim \pi_{\text{dpo}}(\cdot|x)} \left[ r^*(x, y^1) - \widehat{r_{\text{dpo}}}(x, y^1) - r^*(x, y^2) + \widehat{r_{\text{dpo}}}(x, y^2) \right]$$

$$\leq \sqrt{\mathbb{E}_{x \sim \rho} \mathbb{E}_{y^1 \sim \pi^*(\cdot|x), y^2 \sim \pi_{\text{dpo}}(\cdot|x)} \left[ (r^*(x, y^1) - \widehat{r_{\text{dpo}}}(x, y^1) - r^*(x, y^2) + \widehat{r_{\text{dpo}}}(x, y^2))^2 \right]}$$

$$\leq \sqrt{C_{\text{glo}}^2 \mathbb{E}_{x \sim \rho} \mathbb{E}_{y^1, y^2 \sim \pi_{\text{ref}}(\cdot|x)} \left[ (r^*(x, y^1) - \widehat{r_{\text{dpo}}}(x, y^1) - r^*(x, y^2) + \widehat{r_{\text{dpo}}}(x, y^2))^2 \right]},$$

and we can complete the proof by plugging in the error guarantee from Assumption 4.3. $\square$

Note that as the proof suggests, the result holds with the more general reward learning guarantee as in Lemma B.2 – one only needs to be accurate in predicting the relative rewards between response pairs.

## 4.3 Online RL Method Under Partial Coverage

Finally, we contrast the previous negative results in Section 4.1 for offline contrastive-based algorithms to a positive result for online RL-based algorithms, under the partial coverage setting. We will show that in general global coverage is not necessary for RLHF, i.e., it can guarantee performance under partial coverage. In fact, one might still be able to show an impossibility result for RLHF under partial coverage, by reusing the same counterexample as in the previous section (c.r., Proposition 4.1). Concretely, as long as the learned reward $\widehat{r}(y_3) \to \infty$, $\pi_{\text{rlhf}}(y_3)$ will be 1 and thus the reverse KL will be unbounded. However, this is a rather unrealistic scenario, as the construction requires a reward model (e.g., a neural network) to output an unbounded value. Thus this motivates the following assumption:

**Assumption 4.4.** *For all learned reward model $\widehat{r}$ from the reward model class, we have that $\|\widehat{r}\|_\infty \leq R'$.*

At this point, one might ask why a similar assumption is missing for the offline contrastive-based analysis, since in Remark 4.2 we argued that a point-wise learning guarantee is unrealistic but Assumption 4.4 is indeed also a point-wise boundedness assumption. The reason lies in the different construction of the model class $\widehat{r}$ for those algorithms: for DPO and IPO, the reward model is constructed as $\widehat{r_{\text{dpo}}} = \beta \log\left(\frac{\pi}{\pi_{\text{ref}} \cdot Z}\right)$, and there is no natural function class for $\pi$ such that point-wise assumptions such as the one in Remark 4.2 or Assumption 4.4 holds. In contrast, post-processing such as clipping, offline normalization and on-the-fly normalization of rewards is standard in practice, which means the policy will always witness bounded rewards [8, 9, 18, 1] during online RL training (e.g., PPO). As we will show in the following, the difference in the reward function (which is tied to the offline vs. online nature of the algorithms) can explain the different coverage requirement of the algorithms. Note that we use the same in-distribution reward learning assumption for both types of methods.

To relate to Assumption 4.2, we first show that the reverse KL divergence of the RLHF policy is always bounded under Assumption 4.4.

**Lemma 4.1.** *Suppose that Assumption 4.4 holds. Then for any RLHF policy $\pi_{\text{rlhf}}$, we have that*

$$\mathsf{KL}(\pi_{\text{rlhf}} || \pi_{\text{ref}}) := \mathbb{E}_{x \sim \rho}\left[\mathbb{E}_{y \sim \pi_{\text{rlhf}}(\cdot|x)}\left[\log\left(\frac{\pi_{\text{rlhf}}(y \mid x)}{\pi_{\text{ref}}(y \mid x)}\right)\right]\right] \leq \frac{2R'}{\beta}.$$

Then we can show that the RLHF algorithm can guarantee performance under partial coverage:

**Theorem 4.2.** *Suppose that Assumption 4.4 holds. Then for any reference policy $\pi_{\text{ref}}$ for which Assumption 4.2 holds with $\varepsilon_{\text{kl}} = \frac{2R'}{\beta}$, and any RLHF policy $\pi_{\text{rlhf}}$ with $\widehat{r}$ such that (c.r. Assumption 4.3) $\mathbb{E}_{x,y \sim \rho \circ \pi_{\text{ref}}}\left[(r^*(x, y) - \widehat{r}(x, y))^2\right] \leq \varepsilon_{\text{reward}}$, we have*

$$J(\pi^*) - J(\pi_{\text{rlhf}}) \leq O(C_{\varepsilon_{\text{kl}}} \sqrt{\varepsilon_{\text{reward}}}).$$

Conditioned on Lemma 4.1, the proof of this theorem is similar to that of Theorem 4.1 so we defer it to Appendix D. Similar to Theorem 4.1, we note that Theorem 4.2 holds under a weaker reward learning guarantee as in Lemma B.2. We also remark that as long as $\varepsilon_{\text{kl}}$ is finite, $C_{\varepsilon_{\text{kl}}}$ is finite, so the bound is never vacuous. *Since $C_{\varepsilon_{\text{kl}}} \leq C_{\text{glo}}$ for all $\varepsilon_{\text{kl}}$, it indicates the regret bound of RLHF is never worse and can be much better than the regret bound of DPO.* Combining Theorem 4.1 and Theorem 4.2, we complete the separation result between offline contrastive methods and online RL methods.

A natural question at this point could be: can we further relax the local KL-ball coverage condition in Assumption 4.2 to a single-policy coverage condition, i.e., just assuming $\max_{x,y} \pi^*(y|x)/\pi_{\text{ref}}(y|x) \leq C$? Prior work [59] shows that with explicit pessimism, it is possible. However, using pessimism makes the algorithm from [59] not computationally tractable and hard to scale to LLM experiments. Our conjecture is that for the RLHF policy $\pi_{\text{rlhf}}$, it is not possible to achieve meaningful regret under the single policy coverage condition, due to KL not being strong enough to induce pessimism (i.e., bounded KL between $\pi$ and $\pi_{\text{ref}}$ can still imply exponentially large density ratio $\pi/\pi_{\text{ref}}$). Developing a lower bound for $\pi_{\text{rlhf}}$ under single policy coverage in this case can be an interesting future work.

---

**Algorithm 1** Hybrid Preference Optimization (HyPO)

**require** Pretrained LLM $\pi_{\theta_0}$, reference policy $\pi_{\text{ref}}$, offline data $\mathcal{D}$, learning rate $\alpha$, KL coefficient $\lambda$.

1: **for** $t = 1, \ldots, T$ **do**
2:      Sample a minibatch of offline data $D_{\text{off}} := \{x, y^+, y^-\} \sim \mathcal{D}$.
3:      Compute DPO loss $\ell_{\text{dpo}} := \sum_{x,y^+,y^- \in D_{\text{off}}} \log\left(\sigma\left(\beta \log\left(\frac{\pi_{\theta_{t-1}}(y^+|x)}{\pi_{\text{ref}}(y^+|x)}\right) - \beta \log\left(\frac{\pi_{\theta_{t-1}}(y^-|x)}{\pi_{\text{ref}}(y^-|x)}\right)\right)\right)$.
4:      Sample (unlabeled) online data $D_{\text{on}} := \{x, y\}$ where $x \sim \mathcal{D}, y \sim \pi_{\theta_{t-1}}(x)$.
5:      Compute $\ell_{\text{kl}} := \sum_{x,y \in D_{\text{on}}} \log(\pi_{\theta_{t-1}}(y|x)) \cdot \text{sg}\left(\log\left(\frac{(\pi_{\theta_{t-1}}(y|x))}{(\pi_{\text{ref}}(y|x))}\right)\right)$.
6:      Update $\theta_t = \theta_{t-1} + \alpha \cdot \nabla_{\theta_{t-1}}(\ell_{\text{dpo}} - \lambda \ell_{\text{kl}})$.
    **return** $\pi_T$.

---

## 5 Hybrid Preference Optimization: Regularizing Offline Learning with Unlabeled Online Samples

In this section, we will provide a practical algorithm that bridges the gap between the offline contrastive-based algorithms and the online RL-based algorithms. As we see in the previous sections, the difference between the two types of algorithms is their reward model parametrization, and whether to perform online rollouts. In the following, we will show that these two properties are in fact tightly intervened with each other.

Here we will focus on the DPO algorithm. One way to fix the issue of the unbounded reverse KL of DPO (which is caused by the unbounded reward model class) is to consider the following ideal procedure: at the beginning of the algorithm, we first go through the policy class $\Pi$, and then we filter out all the policies such that $\mathsf{KL}(\pi||\pi_{\text{ref}}) \geq \frac{2R'}{\beta}$, where $R'$ is the boundedness of the reward function class for RLHF. Now applying the same analysis of Theorem 4.2, we can show that this revised DPO algorithm can guarantee performance under the partial coverage assumption, because now the Lemma 4.1, a sufficient condition for Theorem 4.2, is explicitly enforced by the constraints. We defer the detailed statement and analysis to Appendix F.1.

However, such a filtering procedure is not possible in practice, but we can instead consider the following constrained optimization problem: we call the definition of DPO loss in Eq. (4), we want to solve

$$\max_{\pi} \ell_{\text{dpo}}(\pi) \quad \text{s.t.} \quad \mathsf{KL}(\pi||\pi_{\text{ref}}) \leq \frac{2R'}{\beta}, \tag{6}$$

using the KKT conditions, we can show that the following Lagrangian form is equivalent to Eq. (6):

$$\max_{\pi} \ell_{\text{dpo}}(\pi) - \lambda \mathsf{KL}(\pi||\pi_{\text{ref}}), \tag{7}$$

where $\lambda$ is the Lagrange multiplier. However, in reality, since we do not know the exact value of $R'$, we can consider setting $\lambda$ to be a hyperparameter. We present the pseudocode in Algorithm 1. Note that due to the reverse KL term, the Hybrid Preference Optimization (HyPO) algorithm optimizes Eq. (7) via both offline and online samples where the offline samples are used for constructing and optimizing $\ell_{\text{dpo}}$ (here $\sigma$ denotes the sigmoid function), and the online samples $y \sim \pi(\cdot \mid x)$ are for KL (i.e., $\ell_{kl}$). Note that regularizing with reverse KL via online samples is widely used in online RLHF (e.g., PPO [40], APA [62], REBEL [18]). Here sg refers to the stop gradient operation, which is a common practice in optimizing reverse KL in the LLM fine-tuning setting [30, 47]. Finally, previous iterative RLHF methods [53] can be interpreted as hybrid methods as well, but they require labeling online samples from an additional reward model while HyPO only requires unlabeled online samples.

**Summarization.** Our first experiment is on the TL;DR dataset [40]. Our experiment setup mostly follows [18]: we use a maximum context length of 512 and a maximum generation length of 53. We use Pythia 1.4B and Pythia 2.8B [5] as the pre-trained model. For the supervised fine-tuning (SFT) model, we train it over 1 epoch of the dataset with human reference responses as labels. We train the reward model on top of the SFT over 1 epoch of preference data. Both HyPO and DPO are trained over 1 epoch of preference data with Low-rank Adaptation (LoRA) [21]. We defer more experiment details in Appendix F.

Table 1: Results on TL;DR dataset. Winrate is evaluated by GPT4 and RM score is from the trained reward model. Experiments are repeated for 3 random seeds. Mean and standard deviation are reported.

| Model size | Algorithm | Winrate ($\uparrow$) | RM score ($\uparrow$) | $\text{KL}(\pi||\pi_{\text{ref}})(\downarrow)$ |
|---|---|---|---|---|
| 1.4B | DPO | 42.17% (2.5%) | 0.16 (0.05) | 44.90 (1.29) |
| | HyPO | **46.44% (2.39%)** | **0.37 (0.05)** | **27.07 (2.34)** |
| 2.8B | DPO | 44.39% (0.4%) | 2.43 (0.10) | 68.95 (3.08) |
| | HyPO | **50.50% (1.89%)** | **2.51 (0.13)** | **48.98 (4.23)** |

Table 2: Results on general chat benchmarks. We evaluate the base model (Meta-Llama-3-8B-Instruct), DPO-fine-tuned model, and HyPO-fine-tuned model.

| Model | MT-Bench | | | AlpacaEval 2.0 | |
|---|---|---|---|---|---|
| | 1st Turn | 2nd Turn | Average | LC Win Rate | Win Rate |
| Meta-Llama-3-8B-Instruct [27] | 8.31 | 7.89 | 8.10 | 26.0 | 25.3 |
| DPO-Llama-3 | 8.08 | 7.41 | 7.75 | 28.4 | 30.9 |
| HyPO-Llama-3 | 8.43 | 7.75 | 8.09 | 30.7 | 32.2 |

We summarize the results in Table 1: HyPO outperforms DPO in terms of GPT4 win-rage and reverse KL. Particularly, the significant reduction in reverse KL implies the impact of including a reverse KL term explicitly into the DPO objective. While comparing with PPO (e.g., Table 1 in [18]), HyPO's performance is still lower in winrate, HyPO does preserve the key advantages of DPO over PPO: we avoid training additional reward model and a value network.

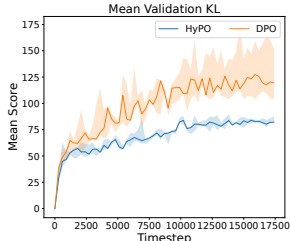

Figure 1: Mean validation reverse KL to the reference policy when DPO and HyPO are trained for 5 epoch on the TL;DR dataset. We repeat the experiment for 3 random seeds and plot the median and the shaded areas denote the min and max over the 3 repetitions.

**General Chat.** In the general chat setting, the model is required to produce a response $y$ given user instruction $x$. We again follow the experiment setup in [18], where we finetune the Meta-Llama-3-8B-Instruct [27] model on the ultrafeedback dataset [14]. Due to the computation constrain, we follow the setup in [18] where we only train the last 4 layers of the network for both HyPO and DPO.

For evaluation, we use the common metrics including AlpacaEval 2.0 [15], MT-bench [61] and Open LLM leaderboard tasks: MMLU [20], GSM8K [13], Arc [12], TruthfulQA [26] and HellaSwag [57]. We provide the results for AlpacaEval and MT-bench in Table 2, and the the results of the remaining tasks can be found in Table 3.

**HyPO Mitigates Overfitting in Contrastive Methods.** Since the offline contrastive based methods only work with a static offline dataset, the overfitting issue has been observed [43]. In our last experiment, we show that HyPO can effectively address the overfitting issue by leveraging the unlabeled online data. We follow the setup of the summarization task with Pythia-2.8B base model. We train DPO and HyPO for 5 epochs respectively, and evaluate on the first 300 data in the validation dataset. We plot the validation KL in Figure 1: we observe that HyPO is better at preventing the deviation from the reference policy caused by overfitting from training on excessive epochs, even though the methods theoretically both have KL regularization to the reference policy.

## 6 Function Approximation Coverage: Can Fine-tuned Policies Extrapolate?

Our final result is a theoretical explanation of the extrapolation behavior of preference fine-tuning algorithms under the global coverage assumption in the function approximation (FA) setting. The extrapolation behavior refers to the phenomenon of RLHF algorithms (e.g., DPO) can improve SFT models despite the fact that during training the policies assign decreasing likelihood to both the preferred and rejected responses (i.e., they must increase the likelihood of responses outside of the training data) [31].

A previous attempt [32] to explain this behavior is based on the assumption that the responses from the reference policy have the same distribution as the *preferred* responses from the dataset, i.e., $y^+ \sim \mu \overset{d}{=} y \sim \pi_{\text{ref}}$. However, as mentioned in Section 3, more realistically, one should assume that

Table 3: Results on Open LLM leaderboard. We evaluate the base model (Meta-Llama-3-8B-Instruct), DPO-fine-tuned model, and HyPO-fine-tuned model.

| Model | MMLU (5-shot) | GSM8K (5-shot) | Arc (25-shot) | TruthfulQA (0-shot) | HellaSwag (10-shot) | Average |
|---|---|---|---|---|---|---|
| Meta-Llama-3-8B-Instruct [27] | 65.68 | 74.91 | 62.12 | 43.88 | 78.76 | 65.07 |
| DPO-Llama-3 | 65.82 | 73.62 | 63.14 | 45.02 | 79.1 | 65.34 |
| HyPO-Llama-3 | 65.74 | 73.84 | 62.71 | 45.55 | 79.74 | 65.51 |

$y \sim \mu \overset{d}{=} y \sim \pi_{\mathsf{ref}}$ since it is more natural to use the reference policy to generate pairs of responses to collect labels; or even more generally by considering $\mathsf{supp}(\mathcal{D}) \subset \mathsf{supp}(\pi_{\mathsf{ref}})$. The latter is common in practice, for example, the dataset is often precollected, or the reference policy might have a small mass on some responses, so with a high probability they are not sampled during the data collection process.

In the following, we illustrate this behavior using linear function approximation:

**Definition 6.1** (Linear function approximation). *Consider the promptless setting with response space $\mathcal{Y}$. For all $y \in \mathcal{Y}$, the ground truth reward $r^*(y) = w^\top \phi(y)$, where $w \in \mathbb{R}^d$ is a universal linear weight vector and $\phi : \mathcal{Y} \to \mathbb{R}^d$ is a d-dimensional feature map. In addition, all policies are parametrized as softmax linear policies, i.e., $\pi(y) \propto \exp(w_\pi^\top \phi(y))$. The feature map $\phi$ is known to the learner, and $w$ is unknown. Finally, let $y^* = \operatorname{argmax}_{y \in \mathcal{Y}} r^*(y)$ be the optimal action.*

In general, we should not expect the offline dataset to contain the optimal action $y^*$ under all situations. We show that thanks to the linear function approximation and the dataset coverage, DPO has hope to extrapolate correctly, i.e., it can increase the model's likelihood of the optimal action while decreasing the likelihood of both the preferred and rejected actions from the offline data:

**Proposition 6.1.** *Under linear function approximation (Definition 6.1), there exists dataset collected from distribution $\mu$ that does not cover $y^*$, i.e., $\mu(y^*) = 0$, but has global coverage in the linear function approximation setting [54]: let $\Sigma_\mu = \mathbb{E}_{y \sim \mu} \phi(y)\phi(y)^\top$, then for all $\pi$, $\mathbb{E}_{y \sim \pi} \|\phi(y)\|^2_{\Sigma_\mu^{-1}} \leq C_\pi$. Then DPO will return a policy $\pi_{\mathsf{dpo}}$ such that $\pi_{\mathsf{dpo}}(y^*) > 0$, but $\pi_{\mathsf{dpo}}(y) \leq \pi_{\mathsf{ref}}(y)$ for all $y$ in the offline data support, i.e., $\mu(y) > 0$.*

We defer the proof to Appendix D.3. The above result shows when the training dataset together with the function approximation allow the learned function to generalize (e.g., learn a function that can predict well on test examples beyond the training data — a property supervised learning can have), algorithms like DPO can extrapolate correctly, i.e., they can push up the likelihood of the optimal responses outside of the training data while pushing down the likelihood of all the responses in the training data. Although it is not possible to show that extrapolation is guaranteed to always happen: suppose in the offline dataset, there is an action $y'$ whose feature $\phi(y')$ is almost identical to $\phi(y^*)$, then in the finite sample case, with small learning error the final policy might increase the probability of $y'$ instead of extrapolate to $y^*$, but in this case the policy is still near optimal. That said, our construction in the proof is general enough to explain non-edge-cases under function approximation.

To validate our theory result, in Appendix E we perform a synthetic experiment on global coverage with linear function approximation.

## 7 Discussion

There are a few limitations of our work: 1) our theoretical analysis only considers the statistical perspective of each algorithm, but we believe our result is complementary to the other work that considers the optimization perspectives [42]. 2) we only conduct experiments on limited models and benchmarks. 3) The experiment result shows that HyPO's performance is still below the one of online RLHF: this might suggest that our theory does not fully explain the benefit of all the components of online RLHF. For example, one hypothesis is that the learn reward function may have better generalization ability. 4) It is not clear that the KL-ball coverage is necessary for online RL-based methods. However, as we discussed, since a bounded reverse KL might still induce exponential error amplification, we conjecture that at least the single policy coverage [58] is not sufficient for online RLHF-based methods that use reverse KL. We believe these limitations lead to several interesting further directions. Finally, our method may not explicitly address the potential hallucinations or toxic behavior of LLMs, which is a common shortcoming of general-purpose fine-tuning algorithms.

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

## A    Additional Related Works

**Extrapolation Behavior of PFT.**    Recent work [56, 31, 32] has observed an interesting effect of the DPO procedure: a simultaneous decrease in the likelihood of both preferred and rejected responses. This behavior is surprising at first glance because one would expect that DPO will increase the likelihood of preferred responses and decrease the likelihood of rejected responses. We provide a rigorous statistical explanation of this behavior and show that this behavior is natural when the offline preference data only contains sub-optimal responses but the function approximation allows DPO to extrapolate and generalize to the correct optimal responses. This highlights the role of function approximation in the success of offline contrastive based methods.

**Coverage.**    We analyze online RLHF and offline contrastive-based methods via the concept of coverage. Coverage measures how well an offline (data) distribution covers the support of the policy of interest, which has been the key technical tool in offline RL [28, 49, 46, 58], offline-online RL [34, 51, 39, 2] and online RL [22, 4, 50]. The data coverage plays an important role in our analysis since both online RLHF and offline contrastive-based methods rely on an offline preference dataset for learning.

## B    Auxiliary Lemmas

**Lemma B.1** (Objective decomposition). *Let $J(\pi)$ be the objective function defined in (1), and for reward function $\hat{r}$, we let*

$$\hat{\pi} \in \operatorname*{argmax}_{\pi} \mathbb{E}_{x \sim \rho}\big[\mathbb{E}_{y \sim \pi(\cdot|x)}[\hat{r}(x,y)] - \beta \mathsf{KL}(\pi(\cdot \mid x)||\pi_{\mathsf{ref}}(\cdot \mid x))\big], \tag{8}$$

*then we have*

$$J(\pi^*) - J(\hat{\pi}) \le \mathbb{E}_{x \sim \rho}\big[\mathbb{E}_{y^1 \sim \pi^*(\cdot|x), y^2 \sim \hat{\pi}(\cdot|x)}\big[r^*(x,y^1) - \hat{r}(x,y^1) - r^*(x,y^2) + \hat{r}(x,y^2)\big]\big].$$

**Proof.** We have

$$J(\pi^*) - J(\hat{\pi})$$
$$= \mathbb{E}_{x \sim \rho}\big[\mathbb{E}_{y \sim \pi^*(\cdot|x)}[r^*(x,y)] - \beta \mathsf{KL}(\pi^*(\cdot \mid x)||\pi_{\mathsf{ref}}(\cdot \mid x))\big] - \mathbb{E}_{x \sim \rho}\big[\mathbb{E}_{y \sim \hat{\pi}(\cdot|x)}[r^*(x,y)] + \beta \mathsf{KL}(\hat{\pi}(\cdot \mid x)||\pi_{\mathsf{ref}}(\cdot \mid x))\big]$$
$$= \mathbb{E}_{x \sim \rho}\big[\mathbb{E}_{y \sim \pi^*(\cdot|x)}[r^*(x,y)] - \beta \mathsf{KL}(\pi^*(\cdot \mid x)||\pi_{\mathsf{ref}}(\cdot \mid x))\big] - \big(\mathbb{E}_{x \sim \rho}\big[\mathbb{E}_{y \sim \hat{\pi}(\cdot|x)}[r^*(x,y)] - \beta \mathsf{KL}(\hat{\pi}(\cdot \mid x)||\pi_{\mathsf{ref}}(\cdot \mid x))\big]\big)$$
$$\quad + \mathbb{E}_{x \sim \rho}\big[\mathbb{E}_{y \sim \hat{\pi}(\cdot|x)}[\hat{r}(x,y)] - \beta \mathsf{KL}(\hat{\pi}(\cdot \mid x)||\pi_{\mathsf{ref}}(\cdot \mid x))\big] - \big(\mathbb{E}_{x \sim \rho}\big[\mathbb{E}_{y \sim \hat{\pi}(\cdot|x)}[\hat{r}(x,y)] - \beta \mathsf{KL}(\hat{\pi}(\cdot \mid x)||\pi_{\mathsf{ref}}(\cdot \mid x))\big]\big)$$
$$\le \mathbb{E}_{x \sim \rho}\big[\mathbb{E}_{y \sim \pi^*(\cdot|x)}[r^*(x,y)] - \beta \mathsf{KL}(\pi^*(\cdot \mid x)||\pi_{\mathsf{ref}}(\cdot \mid x))\big] - \big(\mathbb{E}_{x \sim \rho}\big[\mathbb{E}_{y \sim \pi^*(\cdot|x)}[\hat{r}(x,y)] - \beta \mathsf{KL}(\pi^*(\cdot \mid x)||\pi_{\mathsf{ref}}(\cdot \mid x))\big]\big)$$
$$\quad + \mathbb{E}_{x \sim \rho}\big[\mathbb{E}_{y \sim \hat{\pi}(\cdot|x)}[\hat{r}(x,y)] - \beta \mathsf{KL}(\hat{\pi}(\cdot \mid x)||\pi_{\mathsf{ref}}(\cdot \mid x))\big] - \big(\mathbb{E}_{x \sim \rho}\big[\mathbb{E}_{y \sim \hat{\pi}(\cdot|x)}[r^*(x,y)] - \beta \mathsf{KL}(\hat{\pi}(\cdot \mid x)||\pi_{\mathsf{ref}}(\cdot \mid x))\big]\big)$$
$$= \mathbb{E}_{x \sim \rho}\big[\mathbb{E}_{y \sim \pi^*(\cdot|x)}[r^*(x,y) - \hat{r}(x,y)]\big] - \mathbb{E}_{x \sim \rho}\big[\mathbb{E}_{y \sim \hat{\pi}(\cdot|x)}[r^*(x,y) - \hat{r}(x,y)]\big],$$

where the inequality is due to Eq. (8). To complete the proof, note that

$$\mathbb{E}_{x \sim \rho}\big[\mathbb{E}_{y \sim \pi^*(\cdot|x)}[r^*(x,y) - \hat{r}(x,y)] - \mathbb{E}_{x \sim \rho}\mathbb{E}_{y \sim \hat{\pi}(\cdot|x)}[r^*(x,y) - \hat{r}(x,y)]\big]$$
$$= \mathbb{E}_{x \sim \rho}\big[\mathbb{E}_{y^1 \sim \pi^*(\cdot|x), y^2 \sim \hat{\pi}(\cdot|x)}[r^*(x,y^1) - \hat{r}(x,y^1)]\big] - \mathbb{E}_{x \sim \rho}\big[\mathbb{E}_{y^1 \sim \pi^*(\cdot|x), y^2 \sim \hat{\pi}(\cdot|x)}[r^*(x,y^2) - \hat{r}(x,y^2)]\big]$$
$$= \mathbb{E}_{x \sim \rho}\big[\mathbb{E}_{y^1 \sim \pi^*(\cdot|x), y^2 \sim \hat{\pi}(\cdot|x)}\big[r^*(x,y^1) - \hat{r}(x,y^1) - r^*(x,y^2) + \hat{r}(x,y^2)\big]\big].$$

$\square$

**Lemma B.2** (Lemma C.2 from [9]). *Assume that $r^*$ is bounded, let $\mathcal{R}$ be the reward function class, and Let*

$$\hat{r} = \operatorname*{argmin}_{r \in \mathcal{R}} \hat{\mathbb{E}}_{x, y^+, y^- \sim \mathcal{D}}\left[\log\left(\frac{\exp(r(x,y^+))}{\exp(r(x,y^+)) + \exp(r(x,y^-))}\right)\right],$$

*then we have with probability at least $1 - \delta$ that*

$$\mathbb{E}_{x, y^1, y^2 \sim \mu \circ \pi_{\mathsf{ref}}}\left[\big(r^*(x,y^1) - r^*(x,y^2) - \hat{r}(x,y^1) + \hat{r}(x,y^2)\big)^2\right] \le \frac{c\kappa^2 \log(|\mathcal{R}|/\delta)}{N},$$

*where $\kappa$ measures the non-linearity of the link function, and $c$ is a constant, $N := |\mathcal{D}|$ is the size of the offline dataset.*

# C  Additional Results

## C.1  Results for IPO

In this section we give detailed technical details for IPO, and the negative results for IPO under partial coverage. Recall that the empirical objective of IPO is is $\pi_{\mathsf{ipo}} \in \arg\min_\pi \widehat{\ell_{\mathsf{ipo}}}(\pi)$, where

$$\widehat{\ell_{\mathsf{ipo}}}(\pi) = \widehat{\mathbb{E}}_{x,y^+,y^- \sim \mathcal{D}}\left[\left(\log\left(\frac{\pi(y^+ \mid x)\pi_{\mathsf{ref}}(y^- \mid x)}{\pi(y^- \mid x)\pi_{\mathsf{ref}}(y^+ \mid x)}\right) - \frac{\beta^{-1}}{2}\right)^2\right].$$

The empirical objective is derived from the following population loss

$$\ell_{\mathsf{ipo}}(\pi) = \mathbb{E}_{x,y^1,y^2 \sim \rho \circ \pi_{\mathsf{ref}}}\left[\left(h_\pi\left(y^1,y^2\right) - I\left(y^1,y^2\right)/\beta\right)^2\right], \tag{9}$$

where

$$h_\pi(y^1,y^2) = \log\left(\frac{\pi(y^1)\pi_{\mathsf{ref}}(y_2)}{\pi(y^2)\pi_{\mathsf{ref}}(y_1)}\right),$$

and $I(y^1,y^2)$ is a Bernoulli random variable with parameter $p = p^*(y_1 \succ y_2)$, where here $p^*$ can be any underlying human preference (that is not necessarily parametrized by the Bradley Terry model). To show the negative result, we can make the following learning assumption:

**Assumption C.1** (In distribution guarantee for IPO). *We assume that the returned policy $\pi_{\mathsf{ipo}}$ satisfies that*

$$\pi_{\mathsf{ipo}} = \arg\min_{\pi \in \Pi} \ell_{\mathsf{ipo}}(\pi),$$

*i.e., the returned policy $\pi_{\mathsf{ipo}}$ induces the smallest possible in-distribution error on its population loss.*

With the setup, we can state and prove the formal version of the result:

**Proposition C.1** (Formal version of of Proposition 4.2). *Denote $\pi_{\mathsf{ref}}$ as any reference policy such that Assumption 4.1 breaks. Let $\Pi_{\mathsf{ipo}}$ be the set of IPO returned policies such that Assumption C.1 holds. Then there exists policy $\pi \in \Pi_{\mathsf{ipo}}$ such that $J(\pi) = -\infty$.*

**Proof.** Without loss of generality, we consider a promptless setting, and assume that the response space is $\mathcal{Y} = \{y_1, y_2, y_3\}$. Again without loss of generality, we assume $\pi_{\mathsf{ref}}$ only covers $y_1$ and $y_2$, and thus Assumption 4.1 breaks. Specifically, let $\pi_{\mathsf{ref}}(y_1) = \pi_{\mathsf{ref}}(y_2) = 1/2$. Then we have

$$\pi_{\mathsf{ipo}} = \arg\min_{\pi \in \Pi} \mathbb{E}_{y^1,y^2 \sim \pi_{\mathsf{ref}}}\left[\left(\log\left(\frac{\pi(y^1)}{\pi(y^2)}\right) - I\left(y^1,y^2\right)/\beta\right)^2\right],$$

which gives

$$\log\left(\frac{\pi_{\mathsf{ipo}}(y_1)}{\pi_{\mathsf{ipo}}(y_2)}\right) = p^*(y_1 \succ y_2)/\beta,$$

and thus we have the relation that

$$\pi_{\mathsf{ipo}}(y_1) = \pi_{\mathsf{ipo}}(y_2) \cdot \exp(p^*(y_1 \succ y_2)/\beta).$$

Let $\pi_{\mathsf{ipo}}(y_2) = \alpha \in (0,1]$, then for any $\alpha$ such that $\pi_{\mathsf{ipo}}(y_3) = 1 - (1 + \exp(p^*(y_1 \succ y_2)/\beta))\alpha > 0$, we will have that $\mathsf{KL}(\pi_{\mathsf{ipo}}||\pi_{\mathsf{ref}})$ is unbounded, and thus we complete the proof. $\square$

## C.2  DPO Has Vacuous Forward KL

In this section, we show that in the worst case, the forward KL of DPO is vacuously large. We first see how we can relate the forward KL divergence of the DPO policy with the reward learning guarantee. Consider any DPO policy $\pi_{\mathsf{dpo}}$ and its corresponding reward model $\widehat{r_{\mathsf{dpo}}}$. By construction of the DPO algorithm, we have, for any $x, y$ pair that is covered in the dataset, $\pi_{\mathsf{dpo}}(y \mid x) =$

$\frac{\pi_{\text{ref}}(y|x)\exp(\widehat{r_{\text{dpo}}}(x,y)/\beta)}{Z(x)}$, where $Z(x) = \sum_y \pi_{\text{ref}}(y \mid x)\exp(\widehat{r_{\text{dpo}}}(x,y)/\beta)$. Then the forward KL divergence is

$$\mathbb{E}_{x,y\sim\rho\circ\pi_{\text{ref}}}\left[\log\left(\frac{\pi_{\text{ref}}(y \mid x)}{\pi_{\text{dpo}}(y \mid x)}\right)\right] = \mathbb{E}_{x,y\sim\rho\circ\pi_{\text{ref}}}\left[\log\left(\frac{Z(x)}{\exp(\widehat{r_{\text{dpo}}}(x,y)/\beta)}\right)\right] = \mathbb{E}_{x,y\sim\rho\circ\pi_{\text{ref}}}\left[-\frac{\widehat{r_{\text{dpo}}}(x,y)}{\beta} + \log(Z(x))\right].$$

Although the first term can be easily related to the reward learning guarantee, the second term ($\mathbb{E}_{x\sim\rho}[\log(Z(x))]$) can unfortunately be vacuous without further assumptions. We formalize in the following result:

**Proposition C.2.** *There exist $\pi_{\text{dpo}}$ such that Assumption 4.3 holds, but $\mathsf{KL}(\pi_{\text{ref}}||\pi_{\text{dpo}})$ is arbitrarily large.*

**Proof.** First without loss of generality let us consider that $r^* > 0$. Now suppose there exists $\tilde{y}$ such that $\pi_{\text{ref}}(\tilde{y} \mid x) = \frac{1}{n^4}$ for all $x$, where $n$ will be determined soon. Now suppose that for all $x$, $\widehat{r_{\text{dpo}}}(x,\tilde{y}) - r^*(x,\tilde{y}) = n$ and $\widehat{r_{\text{dpo}}}(x,y) = r^*(x,y)$ for all $y \neq \tilde{y}$. Now we can check that

$$\mathbb{E}_{x\sim\rho}\mathbb{E}_{y\sim\pi_{\text{ref}}(\cdot|x)}\left[(\widehat{r_{\text{dpo}}}(x,y) - r^*(x,y))^2\right] = \frac{1}{n^2},$$

which is diminishing if we take $n$ to be big enough. We can also check that

$$\mathbb{E}_{x\sim\rho}\mathbb{E}_{y\sim\pi_{\text{ref}}(\cdot|x)}\left[-\frac{\widehat{r_{\text{dpo}}}(x,y)}{\beta}\right] \geq -\frac{1}{n^3\beta} - \frac{R}{n^4\beta}$$

and thus the first term will have little impact on the final bound. However, the second term can be lower bounded as follows:

$$\log\left(\sum_y \pi_{\text{ref}}(y \mid x)\exp(\widehat{r}(x,y)/\beta)\right) = \log\left(\sum_y \pi_{\text{ref}}(y \mid x)\exp\left(\frac{r^*(x,y) + \widehat{r}(x,y) - r^*(x,y)}{\beta}\right)\right)$$

$$\geq \log\left(\sum_y \pi_{\text{ref}}(y \mid x)\exp\left(\frac{\widehat{r}(x,y) - r^*(x,y)}{\beta}\right)\right)$$

$$= \log\left(\pi_{\text{ref}}(\tilde{y} \mid x)\exp\left(\frac{\widehat{r}(x,\tilde{y}) - r^*(x,\tilde{y})}{\beta}\right)\right)$$

$$= \frac{n}{\beta} - 4\log(n).$$

Putting everything together, we have

$$\mathsf{KL}(\pi_{\text{ref}}||\pi_{\text{dpo}}) \geq \frac{n}{\beta} - 4\log(n) - \frac{1}{n^3\beta} - \frac{R}{n^4\beta}$$

and since we can take $n$ arbitrarily big we complete the proof. $\qquad\square$

# D  Omitted Proofs

## D.1  Proof of Proposition 4.1

**Proposition D.1** (Restatement of Proposition 4.1). *Denote $\pi_{\text{ref}}$ as any reference policy such that Assumption 4.1 breaks. Let $\Pi_{dpo}$ be the set of DPO returned policies such that Assumption 4.3 holds. Then there exists policy $\pi \in \Pi_{dpo}$ such that $J(\pi) = -\infty$.*

**Proof.** Again as in the proof sketch, without loss of generality, we consider a promptless setting, and assume that the response space is $\mathcal{Y} = \{y_1, y_2, y_3\}$. Again without loss of generality, we assume $\pi_{\text{ref}}$ only covers $y_1$ and $y_2$, and thus Assumption 4.1 breaks. Now consider the optimal policy

$$\pi^*(y) = \frac{\pi_{\text{ref}}(y \mid x)\exp(r^*(y)/\beta)}{Z^*(t)}, \forall y \in \mathcal{Y},$$

where $Z^* = \sum_{y \in \mathcal{Y}} \pi_{\text{ref}}(y \mid x) \exp(r^*(y)/\beta)$, note that by construction $\pi^*(y_3) = 0$.

Then consider the following policy $\pi$ such that

$$\beta \log\left(\frac{\pi(y_1)}{\pi_{\text{ref}}(y_1) \cdot Z^*}\right) = r^*(y_1) - \sqrt{\varepsilon_{\text{dpo}}}, \quad \text{and} \quad \beta \log\left(\frac{\pi(y_2)}{\pi_{\text{ref}}(y_2) \cdot Z^*}\right) = r^*(y_2) - \sqrt{\varepsilon_{\text{dpo}}},$$

Then we have

$$\mathbb{E}_{y \sim \pi_{\text{ref}}}\left[\left(\beta \log\left(\frac{\pi_{\text{dpo}}(y)}{\pi_{\text{ref}}(y \mid x) \cdot Z^*}\right) - r^*(x, y)\right)^2\right] = \varepsilon_{\text{dpo}},$$

thus $\pi$ satisfies Assumption 4.3. Rearranging we can see that $\pi(y_1) < \pi^*(y_1)$ and $\pi(y_2) < \pi^*(y_2)$. Now since $\pi^* = 0$, we have

$$\pi^*(y_1) + \pi^*(y_2) = 1,$$

and combine we get $\pi(y_3) > 0$, which implies $\mathsf{KL}(\pi||\pi_{\text{ref}})$ is unbounded, since $\pi_{\text{ref}}(y_3) = 0$. $\quad\square$

## D.2   Proof of Theorem 4.2

In this section we prove Theorem 4.2:

**Theorem D.1** (Restatement of Theorem 4.2). *Suppose that Assumption 4.4 holds. Then for any reference policy $\pi_{\text{ref}}$ such that Assumption 4.2 holds with $\varepsilon_{\text{kl}} = \frac{2R'}{\beta}$, for any RLHF policy $\pi_{\text{rlhf}}$ with $\widehat{r}$ such that (c.r. Assumption 4.3),*

$$\mathbb{E}_{x,y \sim \rho \circ \pi_{\text{ref}}}\left[(r^*(x, y) - \widehat{r}(x, y))^2\right] \leq \varepsilon_{\text{reward}},$$

*or more generally, the event in Lemma B.2 holds for $\widehat{r}$, we have*

$$J(\pi^*) - J(\pi_{\text{rlhf}}) \leq O(C_{\varepsilon_{\text{kl}}} \sqrt{\varepsilon_{\text{reward}}}).$$

To prove this we first prove the following lemma so we can leverage Assumption 4.2:

**Lemma D.1** (Restatement of Lemma 4.1). *Suppose that Assumption 4.4 holds. Then for any RLHF policy $\pi_{\text{rlhf}}$, we have that*

$$\mathsf{KL}(\pi_{\text{rlhf}}||\pi_{\text{ref}}) := \mathbb{E}_{x \sim \rho} \mathbb{E}_{y \sim \pi_{\text{rlhf}}(\cdot|x)}\left[\log\left(\frac{\pi_{\text{rlhf}}(y \mid x)}{\pi_{\text{ref}}(y \mid x)}\right)\right] \leq \frac{2R'}{\beta}.$$

**Proof.** since we have that $\pi_{\text{rlhf}}(y \mid x) = \frac{\pi_{\text{ref}}(y|x) \exp(\widehat{r}(x,y)/\beta)}{Z(x)}$ for all $x \in \mathsf{supp}(\rho), y \in \mathcal{Y}$, we have

$$\mathsf{KL}(\pi_{\text{rlhf}}||\pi_{\text{ref}}) = \mathbb{E}_{x \sim \rho} \mathbb{E}_{y \sim \pi_{\text{rlhf}}(\cdot|x)}\left[\log\left(\frac{\exp(\widehat{r}(x, y))}{\beta Z(x)}\right)\right] = \mathbb{E}_{x \sim \rho} \mathbb{E}_{y \sim \pi_{\text{rlhf}}(\cdot|x)}\left[\frac{\widehat{r}(x, y)}{\beta} - \log(Z(x))\right].$$

Plugging in the definition of $Z(x)$ we get

$$\log(Z(x)) = \log\left(\mathbb{E}_{y \sim \pi_{\text{ref}}(\cdot|x)}\left[\exp\left(\frac{\widehat{r}(x, y)}{\beta}\right)\right]\right) \geq \mathbb{E}_{y \sim \pi_{\text{ref}}(\cdot|x)}\left[\frac{\widehat{r}(x, y)}{\beta}\right]$$

due to Jensen's inequality. Thus we have

$$\mathsf{KL}(\pi_{\text{rlhf}}||\pi_{\text{ref}}) \leq \mathbb{E}_{x \sim \rho} \mathbb{E}_{y \sim \pi_{\text{rlhf}}(\cdot|x)}\left[\frac{\widehat{r}(x, y)}{\beta}\right] - \mathbb{E}_{x \sim \rho} \mathbb{E}_{y \sim \pi_{\text{rlhf}}(\cdot|x)}\left[\frac{\widehat{r}(x, y)}{\beta}\right] \leq \frac{2R'}{\beta}.$$

$\quad\square$

Now with Lemma 4.1, we can prove Theorem 4.2:

**Proof.** By Lemma B.1, we have

$$J(\pi^*) - J(\pi_{\mathsf{rlhf}})$$

$$\leq \mathbb{E}_{x\sim\rho}\mathbb{E}_{y^1\sim\pi^*(\cdot|x),y^2\sim\pi_{\mathsf{rlhf}}(\cdot|x)}\left[r^*(x,y^1) - \widehat{r}(x,y^1) - r^*(x,y^2) + \widehat{r}(x,y^2)\right]$$

$$\leq \sqrt{\mathbb{E}_{x\sim\rho}\mathbb{E}_{y^1\sim\pi^*(\cdot|x),y^2\sim\pi_{\mathsf{rlhf}}(\cdot|x)}\left[(r^*(x,y^1) - \widehat{r}(x,y^1) - r^*(x,y^2) + \widehat{r}(x,y^2))^2\right]}$$

$$\leq \sqrt{C_{\mathsf{glo}}^2\mathbb{E}_{x\sim\rho}\mathbb{E}_{y^1,y^2\sim\pi_{\mathsf{ref}}(\cdot|x)}\left[(r^*(x,y^1) - \widehat{r}(x,y^1) - r^*(x,y^2) + \widehat{r}(x,y^2))^2\right]}$$

(Lemma 4.1 and Assumption 4.2)

$$\leq C\sqrt{\varepsilon_{\mathsf{reward}}}.$$

(Lemma B.2)

$\square$

### D.3 Proof of Proposition 6.1

**Proposition D.2** (Restatement of Proposition 6.1). *Under linear function approximation (Definition 6.1), there exists dataset collected from distribution $\mu$ that does not cover $y^*$, i.e., $\mu(y^*) = 0$, but has global coverage in the linear function approximation setting [54]: let $\Sigma_\mu = \mathbb{E}_{y\sim\mu}\phi(y)\phi(y)^\top$, then for all $\pi$, $\mathbb{E}_{y\sim\pi}\|\phi(y)\|_{\Sigma_\mu^{-1}}^2 \leq C_\pi$. Then DPO will return a policy $\pi_{\mathsf{dpo}}$ such that $\pi_{\mathsf{dpo}}(y^*) > 0$, but $\pi_{\mathsf{dpo}}(y) \leq \pi_{\mathsf{ref}}(y)$ for all $y$ in the offline data support, i.e., $\mu(y) > 0$.*

**Proof.** Consider a response space $\mathcal{Y} = \{y_1, y_2, y_3\}$, with $\phi(y_1) = [1,0], \phi(y_2) = [1/2, 1/2], \phi(y_3) = [0,1]$ Let $w_{\mathsf{ref}} = [1,1]$, then we have $\pi_{\mathsf{ref}}(y_i) = 1/3, \forall i \in \{1,2,3\}$. Let the ground truth reward function $r^*(y) = [10,1]^\top\phi(y)$, and suppose $\mathsf{supp}(\mu) = \{y_1, y_2\}$, i.e., the data only covers $y_1$ and $y_2$, and $\mu(y_3) = 0$. And as always, the preference is based on the ground truth reward function under the Bradley-Terry model.

We can first check that the data distribution indeed has global coverage: for all $\pi$ we have $\mathbb{E}_{y\sim\pi}\|\phi(y)\|_{\Sigma_\mu^{-1}}^2 \leq C_\pi$. If we parameterize $\widehat{r}(y) = \widehat{w}^\top\phi(y)$ (or in case of DPO, we can still check and see that $\widehat{r_{\mathsf{dpo}}}(y) = \widehat{w_{\mathsf{dpo}}}^\top\phi(y)$ because of the softmax linear parametrization of the policies), for either direct reward learning or DPO, we can have the learned reward function $\widehat{r}(y) = [10,1]^\top\phi(y) + c$, where $c$ is the constant reward shift (c.r. Eq. (5)). Then a simple calculation (by $\pi(y) \propto \pi_{\mathsf{ref}}(y)\exp(\widehat{r}(y)/\beta)$) shows that, as long as $c$ is small enough, the policies will decrease the likelihood of $y_1$ and $y_2$ and increase the likelihood of $y_3$. $\square$

## E  Synthetic experiment for extrapolation

### E.1  Extrapolation with function approximation

We first describe our experiment setup. We consider linear function approximation setting where we have 100 responses ($|\mathcal{Y}| = 100$). We consider a 16-dimensional feature vector $\phi : \mathcal{Y} \to \mathbb{R}^{16}$, and we generate $\phi(y)$ by simply sampling 99 random 16-dimensional vectors where the $\ell_1$ norm of each vector is 1. We add one final $\phi(y) = [1,0,0,\dots]$.

We construct the implicit human reward $r^*(y) = w^{*\top}\phi(y)$, where $w^* = [5,...]$, and the rest of the entries are sampled from Unif(-2,2).

We parametrize the policies as softmax linear policies, i.e., we parametrize each policy $\pi$ with $w^\pi \in \mathbb{R}^{16}$ such that $\pi(y) = \frac{w^{\pi\top}\phi(y)}{\sum_{y\in\mathcal{Y}}w^{\pi\top}\phi(y)}$. One can check in this formulation the implicit reward in DPO ($\widehat{r_{\mathsf{dpo}}}$) is linear in $\phi$.

We generate 10000 preference pairs, according to the BT model under $r^*$, for the first 50 responses. We checked that the first responses indeed span $\mathbb{R}^{16}$. Thus the offline data has global coverage in linear function approximation setting.

For on-policy RL methods, we first train a reward model. Then we simply perform gradient descent on the KL-regularized bandit loss (we assume $\pi_{\mathsf{ref}}$ is uniform). For DPO, we simply perform SGD

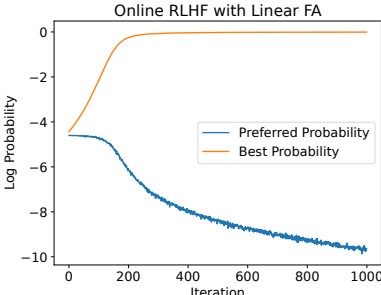
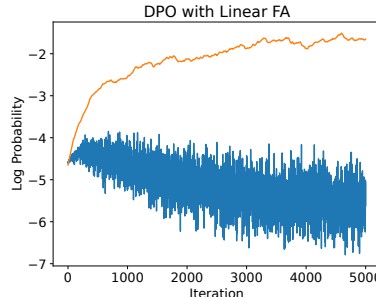

Figure 2: Extrapolation behavior of Online RL method and DPO under linear function approximation. We plot the mean log probability of the preferred responses and the log probability of the best response, which is unseen in the training data. We see that both algorithms correctly assigns increasing probability to the best response.

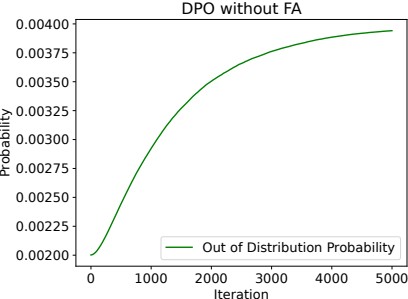

Figure 3: Extrapolation behavior of DPO without function approximation. We plot the average probability of out-of-distribution responses along the training and DPO assigns increasing probability to out-of-distribution responses.

on the offline preference dataset. We track two qualities over the training: the mean log probability of a random subset of preferred responses, and the log probability of best response $\phi(y) = [1, 0, 0, \dots]$. We plot the results in Figure 2. We observe that both methods have the extrapolation behavior – the probability of preferred responses decays but the probability of the optimal response goes up.

### E.2 Extrapolation without function approximation

Now we describe the setting where function approximation fails, and this reduces to a Multi-arm bandit setting. We set $|\mathcal{Y}| = 500$, and the offline data only covers the first half of the responses. The $r^*(y)$ is set by sampling from Unif(-10,10), and we generate 10000 offline samples by uniformly sample pairs of responses from the first half of the response space, and then label them with BT model under $r^*$. We train DPO with 5000 iterations, and plot the mean probability of the responses *outside* of the data support in Figure 3: we observe that the mean probability of the out-of-distribution responses are increasing, however, this could be an undesirable behavior because the reward of the out-of-distribution responses could be arbitrarily bad.

## F  Details of Section 5

### F.1  Theoretical guarantee

In this section, we consider the constrained optimization version of HyPO (Eq. (6)). Note that the reward function class is identical to DPO, i.e., $\mathcal{R}_{\text{hypo}} = \left\{ \beta \log\left( \frac{\pi(y|x)}{\pi_{\text{ref}}(y|x)Z(x)} \right) \mid \pi \in \Pi \right\}$, where $Z(x)$ is the partition function. Then for each output policy $\pi_{\text{hypo}}$, we can denote its implicit reward function

$\widehat{r_{\mathsf{hypo}}}(x, y) := \beta \frac{\pi_{\mathsf{hypo}}(y|x)}{\pi_{\mathsf{ref}}(y|x) \cdot Z(x)}$, and similarly to Theorem 4.2, we can obtain the following guarantee in the partial coverage condition:

**Theorem F.1.** *For any reference policy $\pi_{\mathsf{ref}}$ such that Assumption 4.2 holds with $\varepsilon_{\mathsf{kl}} = \frac{2R'}{\beta}$, for any HyPO policy $\pi_{\mathsf{hypo}}$ such that the event in Lemma B.2 holds, i.e.,*

$$\mathbb{E}_{x, y^1, y^2 \sim \mu \circ \pi_{\mathsf{ref}}} \left[ \left( r^*(x, y^1) - r^*(x, y^2) - \widehat{r_{\mathsf{hypo}}}(x, y^1) + \widehat{r_{\mathsf{hypo}}}(x, y^2) \right)^2 \right] \leq \varepsilon_{\mathsf{hypo}},$$

*we have*

$$J(\pi^*) - J(\pi_{\mathsf{hypo}}) \leq O(C_{\varepsilon_{\mathsf{kl}}} \sqrt{\varepsilon_{\mathsf{hypo}}}).$$

**Proof.** The proof mostly follows the proof of Theorem 4.2. It remains to show the following two properties:

1) Note that Theorem 4.2 requires Assumption 4.4, which does not hold for $\widehat{r_{\mathsf{hypo}}}$ (note that $\widehat{r_{\mathsf{hypo}}}$ is only bounded under $\rho$, but not for all $x$), but we only use it to prove the sufficient condition in Lemma 4.1, which is satisfied by the constraint of HyPO.

2) We need to check that the premise of Lemma B.1 holds, i.e.,

$$\pi_{\mathsf{hypo}} \in \operatorname*{argmax}_{\pi} \mathbb{E}_{x \sim \rho} \left[ \mathbb{E}_{y \sim \pi(\cdot|x)}[\widehat{r_{\mathsf{hypo}}}(x, y)] - \beta \mathsf{KL}(\pi(\cdot \mid x) || \pi_{\mathsf{ref}}(\cdot \mid x)) \right],$$

note that with the reparametrization between $\pi_{\mathsf{hypo}}$ and $\widehat{r_{\mathsf{hypo}}}$, $\pi_{\mathsf{hypo}}$ is always among the minimizer of the *unconstrained* policy set, so we can still invoke Lemma B.2. The rest of the proof now follows the proof of Theorem 4.2 so we omit the details. $\square$

Finally, we remark the connection to the negative result of DPO, i.e, Proposition 4.1: note that given $\mathsf{KL}(\pi_{\mathsf{hypo}} || \pi_{\mathsf{ref}}) \leq \infty$, we have that for all $x$ such that $\rho(x) > 0$, we have for all $y$, $\beta \log \left( \frac{\pi_{\mathsf{hypo}}(y|x)}{\pi_{\mathsf{ref}}(y|x)} \right) < \infty$, (again with the convention that $\frac{0}{0} = 0$), which breaks the construction of Proposition 4.1.

## F.2 Experiment details

### F.2.1 Summarization

In this section, we provide more details of our summarization experiment. We use the Pythia 1.4B and 2.8B model [5] with hugging face model cards: EleutherAI/pythia-1.4b-deduped and EleutherAI/pythia-2.8b-deduped. The TL;DR dataset is available at https://github.com/openai/summarize-from-feedback. The human reference dataset contains 117k training, 6.45K validation and 6.55K testing data. The preference dataset contains 92.9K training and 83.8K validation data. The reward evaluation and KL computation is performed on the whole validation data of the reference dataset. The GPT winrate is computed on a subset of 600 samples from the validation data. The GPT API checkpoint we use is gpt-4-0613. We follow the standard prompt for the winrate evaluation (e.g., see Appendix D.3 of [18]). Below we provide the hyperparameter for HyPO and DPO. Note that to optmize the online KL, we use Reinforce with Leave One Out (RLOO) [24] with two generations per prompt ($k = 2$) and optimize trajectory-level KL.

For our experiment, we run on a cluster of mixture of Nvidia A6000 and L40 GPUs with 48 GB VRAM. We use 4 GPUs in parallel for training, and for DPO the experiment time varies from 1 hour to 2 hours to finish, and for HyPO the time varies between 4 hours to 5 hours.


<div>

Table 4: RM/SFT hyperparameters.

| | |
|---|---|
| Learning rate | 3e-6 |
| Batch size | 64 |
| Learning rate scheduler | cosine |
| Optimizer | Adamw |
| LoRA | False |

</div>
<div>

Table 5: DPO hyperparameters.

| | |
|---|---|
| Learning rate | 3e-6 |
| Batch size | 64 |
| Learning rate scheduler | cosine |
| Optimizer | Adamw |
| $\beta$ | 0.05 |

</div>
</div>

Table 6: HyPO hyperparameters.

| Learning rate | 3e-6 |
|---|---|
| Batch size | 64 |
| Learning rate scheduler | cosine |
| Optimizer | Adamw |
| $\beta$ | 0.05 |
| $\lambda$ | 0.0001 |
| RLOO $k$ | 2 |

Table 7: Lora configurations.

| $r$ | 1024 |
|---|---|
| $\alpha$ | 2048 |
| Dropout | 0 |

### F.2.2 General Chat

For the base model of general chat experiments, we use Llama3-8B-Instruct [27] with hugging face model card: meta-llama/Meta-Llama-3-8B-Instruct. The dataset card of the Ultrafeedback dataset [14] is HuggingFaceH4/ultrafeedback_binarized. In addition to the KL penalty, in the general chat task we add an additional length penalty, and the online penalty of a generation $y$ with context $x$ becomes $\log\left(\frac{\pi(y|x)}{\pi_{\text{ref}}(y|x)}\right) + \alpha|y|$. We summarize the hyperparameter of each baseline below.

We run the general chat experiment on a node of 8 Nvidia A100 80GB GPUs. DPO takes 3 hours to train one epoch while HyPO takes 18 hours to train one epoch.

Table 8: HyPO hyperparameters.

| Learning rate | 3e-6 |
|---|---|
| Batch size | 8 |
| Learning rate scheduler | linear |
| Optimizer | Adamw |
| $\beta$ | 0.05 |
| $\lambda$ | 0.0002 |
| RLOO $k$ | 2 |
| $\alpha$ | 0.02 |

Table 9: DPO hyperparameters.

| Learning rate | 3e-6 |
|---|---|
| Batch size | 8 |
| Learning rate scheduler | linear |
| Optimizer | Adamw |
| $\beta$ | 0.05 |

