# OpenReview forum: "The Importance of Online Data: Understanding Preference Fine-tuning via Coverage"
_NeurIPS.cc/2024/Conference — NeurIPS 2024 poster_

### Official Review · Reviewer_MDLu · 2024-07-08

**Soundness:** 3
**Presentation:** 3
**Contribution:** 3
**Rating:** 6
**Confidence:** 4

**Summary:**

This paper focuses on the optimization and learning methods for ``online'' RLHF and contrastive offline methods (DPO, IPO). The authors aim to understand the separation between these two type of methods, where they are different in terms of whether new responses can be sampled or not. The authors state that the difference in the reward parameterization is key to the separation and propose global coverage and local coverage to capture such a difference.

The theoretical insights also motivate a novel approach called Hybrid Preference Optimization (HyPO) that combines offline data with online samples, where the online samples are used to control the KL divergence. Empirical results are provided to verify the effectiveness of the proposed methods.

**Strengths:**

- The authors study the theory of RLHF under the KL-regularized target, which is more close to the practice, as compared to many previous works using the non-regularized reward;
- The notion of local coverage condition is novel in the literature and is very natural in the analysis of KL-regularized target. I appreciate the paper writing at the beginning of section 4, which is easy to follow and informative.
- Building open the notions of local convergence and global convergence, the authors show a clear separation between the offline algorithms like DPO and online RL-based algorithms. This aligns with the recent observations that the online algorithms outperform their offline counterparts with a large margin.
- This paper takes a step further to study the differences between the DPO and RLHF. While the original DPO paper states that the learning of DPO is equivalent to the RLHF, the empirical results do not align with this. The discussion related to the parametrization of reward for assumption 4.4 explains such a difference in practice.
- Overall, I feel that the story of this paper is complete. Lemma 4.1 and the discussion  around assumption 4.4 clearly show that the offline algorithms can search for the policy with a large KL (possibly due to the parameterization of reward). This not only aligns with the separation between global coverage condition and the local coverage condition, but also motivates the practical algorithmic design to explicitly control the KL divergence.

**Weaknesses:**

This is not a weakness but some clarification on the terminology. In the literature of RL theory, particularly for the preference learning paper, the learning (online exploration) is mentioned with querying the human preference so as to learn the $r^*$. In the setup of this paper, the online data is only used to compute the KL loss without querying the human feedback. Therefore, it is more related to an intermediate setting that we use the offline preference dataset to construct a proxy reward function and we are allowed to query the model to get new responses but not the human feedback. See the RSO [1], some discussions in [2] for proposing multi-step RSO, and some discussions in [3].

[1] Statistical rejection sampling improves preference optimization.
[2] Iterative Preference Learning from Human Feedback: Bridging Theory and Practice for RLHF under KL-constraint
[3] Preference Fine-Tuning of LLMs Should Leverage Suboptimal, On-Policy Data.

Then, it would be interesting to see the empirical results in terms of the ONLINE setting where we can query the new response and the new human preference signals because a lot recent works show that even with the DPO, the online framework outperforms the original one with a large margin. Moreover, this is standard in the PPO literature like Chat-GPT, Claude, and LLaMA2.

[4] Training language models to follow instructions with human feedback
[5] Training a Helpful and Harmless Assistant with Reinforcement Learning from Human Feedback
[6] Llama 2: Open Foundation and Fine-Tuned Chat Models

**Questions:**

see weakness part

**Limitations:**

yes

---

> ### Author Rebuttal · Authors · 2024-08-06
>
> We thank the reviewer for their insightful comments and we address the reviewer's comment below:
>
> > Discussion on online data.
>
> Thank you for pointing out the subtlety of the terminologies. We agree that HyPO indeed does not query any additional information from $r^\ast$, which is different from the papers that the reviewer brought up and some related works that we discussed in the related work section. We agree with the reviewer that HyPO is different from the other online RLHF methods because we only have unlabeled online samples (i.e., the data is not labeled with a reward model). One can see it as a monte-carlo estimation of the reverse KL. We will make this distinction clear with additional discussion with the works mentioned by the reviewer.
>
> We also believe that investigating the standard online setting has important empirical values. In our setting, if we use the reward model that is trained on the same offline dataset to label the online data, then theoretically this is equivalent to HyPO which avoids storing and querying the extra reward model. In Fig.1 in the supplemental material of the rebuttal, we compare the memory cost between DPO, HyPO and PPO (which requires the additional reward model), and we show that HyPO is almost as memory efficient as DPO, where PPO requires more than twice memory than DPO and HyPO (although it also requires to store a value function, but that still indicates the additional memory overhead of the reward model). That said, in the case where one has additional GPU memory, many recent and contemporary works [1,2 and many others] showed that empirically, such an online DPO (or iterative DPO) method indeed greatly improves over the offline DPO results, indicating the effectiveness of online reward information. We will make sure to add additional discussion on this topic in our final version.
>
> [1] RLHF Workflow: From Reward Modeling to Online RLHF. 2024
>
> [2] Exploratory Preference Optimization: Harnessing Implicit Q*-Approximation for Sample-Efficient RLHF. 2024
>
>
> We also want to mention that in the general response section, we performed additional large-scale experiment on HyPO on finetuning Llama 3 8B instruct model, and we hope this additional empirical result can further demonstrate the effectiveness of online unlabeled data alone.

---

> > ### Comment · Reviewer_MDLu · 2024-08-07
> > **Thanks for the responses**
> >
> > For DPO-like algorithm, the memory efficiency is not very important because we do not train a value network. The bottleneck of the PPO algorithm is from the challenges of training policy model and value model simultaneously. In particular, the reward model, and reference model to compute the KL divergence can be served with a remote server.
> >
> > Personally, I believe that the coverage condition is hard to satisfy in the context of LLMs and online exploration and online iterative training are standard in making the state-of-the-art models like llama3.1, gemini, Claude, and gpt according to their technical report. In these settings, the reward models are typically trained on a much larger and more diverse preference dataset (compared to the training set of DPO), and is further combined with some human-written rules, and llm-as-a-judge, which are a proxy of the r^* or P^* (see llama 3.1 report or [1] RLHF Workflow: From Reward Modeling to Online RLHF. 2024 for examples). This setting is different from the online setting considered in this paper. I believe that further study in this setting can be promising as future work.

---

> > > ### Author Response · Authors · 2024-08-12
> > > **Reply to reviewer MDLu**
> > >
> > > We thank the reviewer for their insightful comments. We agree with the reviewer that if we have a reward model which is trained on a more diverse preference dataset (which has a better coverage condition), then it will be beneficial for hypo to query labeled online samples which will provably improve upon our current setting where we do not acquire additional reward information with the unlabelled online samples. As the reviewer points out, a refined study on such iterative/online methods is indeed an important future direction. We are happy to see that the reviewer agrees that the coverage condition is hard to satisfy in the LLMs setting and thus in this work we argue that algorithms that require weaker coverage condition can have better empirical performance, and thus one of the advantage of the iterative methods is to further relax the coverage condition through online exploration. Again we will make sure to distinguish the iterative/online setting from the setting from the current paper in the revised version, with additional discussion with the iterative/online methods including but not limited to the works mentioned in our discussion.

---

### Official Review · Reviewer_yufq · 2024-07-10

**Soundness:** 3
**Presentation:** 3
**Contribution:** 3
**Rating:** 6
**Confidence:** 3

**Summary:**

This work considers the statistical separation between contrastive algorithms (DPO and IPO) and RLHF. It proves that DPO/IPO requires global convergence assumption which is in general a very strong assumption while on the other hand RLHF only requires local coverage. This separation stems from the explicit KL constraint in the objective of RLHF while the implicit reward in contrastive algorithms can results in unbounded KL. Given the observation, a hybrid (offline + online) algorithm is proposed: the preference optimization remains offline DPO however it draws online samples to enforce KL constraints on DPO policy, recovering the statistical guarantee of RLHF while attains the practical computational strengths (w/o value and reward models) of DPO.

**Strengths:**

- Well written and easy to follow; understanding DPO and RLHF is important towards better preference optimization algorithms.

- Good technical qualities and offer viable insights into DPO.

- The proposed hybrid algorithm recovers the statistical guarantee of RLHF while keeps the practical computational strengths of DPO (w/o using value and reward models).

**Weaknesses:**

I only have a couple of minor comments:

- Second sentence of L562 should be appended to L231 to make the proof sketch immediately clear.

- Theorem E.1 could be relocated to Section 5 to be more self-contained.

**Questions:**

- I might have overlooked, what is the intuition behind online KL evaluation? Is it possible to enforce such KL constraint purely in offline setting?

**Limitations:**

Overall, I am convinced this is a good submission with good technical quality. However, as I did not extensively follow the theoretical results on preference optimization, I am not able to comment on the novelty of this work, hence I give confidence score of 3.

It would also be beneficial if the authors could create a table summarizing previous works and highlighting their contributions in comparison to prior research.

---

> ### Author Rebuttal · Authors · 2024-08-06
>
> We thank the reviewer for the postive feedback and we hope our rebuttal can be helpful to further demonstrate our contribution:
>
> > Second sentence of L562 should be appended to L231 to make the proof sketch immediately clear. Theorem E.1 could be relocated to Section 5 to be more self-contained.
>
> Thank you for your suggestion for improving the writing of the paper. We will incorporate these changes in the final version which allows additional pages.
>
> > What is the intuition behind online KL evaluation? Is it possible to enforce such KL constraint purely in offline setting?
>
> The reason for the additional online KL is exactly because purely offline methods can not enforce the KL constraints with only the offline data (Proposition 4.1). The intuition behind Proposition 4.1 is that, since DPO is modeling reward in the form of log ratio between policies, even though it is accurate over the offline data, the behavior of the policy outside the offline data is uncontrolled, which can not enforce the KL constraint, because in reverse KL is measured under the trained policy. In order to prove that offline data alone suffices to control the KL, one has to assume a condition similar to that the offline data covers all possible sequences of tokens (Theorem 4.1), which is unrealistic in most situations.
>
> > Comparison to prior research.
>
> Thank you for your suggestions and we hope our rebuttal can help clarify our theoretical contributions. We will provide a Table comparing with the relevant previous theoretical RLHF works that we mentioned in the related work section in the final version.

---

### Official Review · Reviewer_ewpX · 2024-07-13

**Soundness:** 3
**Presentation:** 3
**Contribution:** 2
**Rating:** 6
**Confidence:** 3

**Summary:**

The paper focuses on the paradigm of fine-tuning large language models (LLMs) using human preference data. It delves into two primary techniques: online reinforcement learning (RL) and offline contrastive methods. The authors challenge the previous notion of these techniques being equivalent by conducting a theoretical analysis through the lens of dataset coverage. They introduce the concepts of global and partial coverage conditions, proving that the former is necessary for offline methods using forward KL like DPO to converge to the optimal policy, while the latter is sufficient for online RL methods using reverse KL. The paper proposes a hybrid preference optimization (HyPO) algorithm, demonstrating its empirical superiority over DPO while maintaining some computational efficiency. The authors also provide a coverage-based explanation for why RL and offline contrastive methods might decrease the probability of preferred responses.

**Strengths:**

1. The paper provides a rigorous mathematical analysis of the different conditions under which offline and online methods have provable performance guarantees, contributing to the theoretical foundation of preference learning in RL.

2. The authors introduce the HyPO algorithm and support their theoretical findings with empirical results, demonstrating the effectiveness of HyPO on the TL;DR dataset.

**Weaknesses:**

1. The experiment is not sufficient. Since it is done with Pythia 1.4B and the TL;DR dataset, it is unclear if the proposed method is still valid on larger models and other datasets. While theoretically interesting, it is unclear if using reverse KL instead of forward KL can lead to a significant performance gain in practice.

2. The proposed HyPO method only uses online samples to calculate the KL loss rather than to collect new preference feedback. While simpler, it may fail to fully leverage the benefits of online methods.

**Questions:**

See the Weaknesses section.

**Limitations:**

The authors discussed the limitations well in the paper.

---

> ### Author Rebuttal · Authors · 2024-08-06
>
> We thank reviewer for the helpful comments and we address them below:
>
> > Insufficient experiments.
>
> We agree that experimenting on larger model sizes and additional dataset is important to demonstrate the effectiveness of HyPO and validate our theory. In the supplementary material for the rebuttal, we include the experiment for finetuning Llama 3 8B-instruct model on UltraFeedback dataset, and provide comparison with DPO on MT-Bench, AlpacaEval, and Open LLM Leaderboard (Table 2 and 3), the standard large-scale finetuning setup. We refer the reviewer to the common response section for more details for these results.
>
> > The proposed HyPO method only uses online samples to calculate the KL loss rather than to collect new preference feedback. While simpler, it may fail to fully leverage the benefits of online methods.
>
> We believe that it is beneficial to obtain additional online information such as reward information when the resource permits. However, gathering online reward information either requires additional human labellers or another reward model, which the latter reduces to online RLHF which requires additional computational resources. We performed a controlled experiment on the computational comparison between DPO, HyPO and PPO in Fig 1 of the supplementary material, which demonstrates the computational advantage of not using a reward model. On the other hand, if the reward model is trained on the same offline dataset, then theoretically one does not require the online reward since it does not introduce new information. (However, in practice people also observe that using reward model indeed improves performance even though it does not introduce new information, which as we pointed out in the limitation section is an interesting future theory direction to pursue).

---

### Official Review · Reviewer_YCrR · 2024-07-19

**Soundness:** 4
**Presentation:** 4
**Contribution:** 3
**Rating:** 7
**Confidence:** 4

**Summary:**

This paper studies learning from human preference feedback for aligning large language models (LLMs). Many existing works share the same observation that offline alignment methods such as DPO underperforms their online counterpart such as PPO. But this phenomenon has not been well understood. This paper studies this difference through the lens of coverage. It unveils that the KL constrain in offline alignment methods can be violated due to incomplete coverage and thus offline methods may find bad solutions. Inspired by this insight, the authors propose a new method, hybrid preference optimization (HyPO), which combines the offline preference update with the online KL regularization. Experiments on the summarization task demonstrate that HyPO is better than DPO. This paper also sheds light on a counterintuitive observation that DPO often decreases the likelihood of preferred responses through an illustrative example.

**Strengths:**

This work addresses an important matter and provides insights on a widely discussed question - why do offline alignment methods often underperform online methods? The paper is well written and easy to follow. The theoretical claims are sensible and easy to understand. I believe the community will benefit a lot from this work.

**Weaknesses:**

The computational cost of the proposed HyPO algorithm is not sufficiently discussed. Usually the most computational cost in online alignment methods comes from the sampling process of LLMs. In practice, the computational cost plays an important role in comparing algorithms. Thus it will provide useful guidance if the computational cost is well discussed in the paper.

The discussion in Section 6 is bit handwavy. While Example 6.1 provides good intuition, a more rigorous analysis is needed to justify the claim.

One minor suggestion on presentation: It will be good to provide the PPO numbers in Table 1 to help the readers compare HyPO to PPO more easily.

**Questions:**

In the TL;DR experiment, I am curious to know what happens if the authors also train HyPO for 4 epochs. Would it perform better than PPO?

**Limitations:**

This paper includes a fairly thorough discussion on the limitations at the end. I would like to add one point: the analysis in this work only focuses on the solution space. It explains why offline methods may find bad solutions, but doesn't explain why they often find the bad ones in practice. As far as I know, people often use a very low learning rate for DPO in practice. For example, the original paper used 1E-6. One possible reason for such low learning rates is that it compensates the ineffective KL regularization. Despite such efforts, DPO still finds bad solutions. It will be insightful if we get a better understanding of why bad solutions seem inevitable.

---

> ### Author Rebuttal · Authors · 2024-08-06
>
> We thank the reviewer for the constructive feedback and we address the reviewer's comments below:
>
> > Computational cost analysis.
>
> We thank the reviewer for pointing this out. Indeed the computational analysis is important in comparing finetuning methods. We had a short discussion at the end of Appendix E.2 in the submission but the runtime comparison is performed across a mixed types of computational resource. Towards a rigorous analysis of the computational comparison, in the supplemental material for the rebuttal we include the results of the computational comparison in Fig. 1 – we fix the computational resource on the same node with 8 A6000 GPUs, and we run each algorithm for a batch of 512 prompts, averaged over 50 batches. We use the reported hyperparameter for each algorithm (DPO and HyPO’s hyperparameter can be found in Table 3 and 4 in the submission, and PPO’s hyperparameter is in Table 5 of the supplementary material).
>
> > The discussion in Section 6 is bit handwavy. While Example 6.1 provides good intuition, a more rigorous analysis is needed to justify the claim.
>
> Thank you for the suggestion. Yes indeed example 6.1 can be turned into a rigorous proof which states (in high level): under linear function approximation, as long as the offline data has global coverage in terms of relative condition number (instead of density ratio), and the optimal response is not in the offline dataset, then the extrapolation behavior of preference learning algorithms will provably happen. We will update this in the final version.
>
> >  Adding PPO numbers.
>
> We adopt the PPO result from [1], which uses the https://github.com/vwxyzjn/summarize_from_feedback_details codebase. We used the same codebase to report DPO performance and implement HyPO. We include the PPO performance in Table.1 in the supplementary material of the rebuttal. The hyperparameter for PPO we use in the computation analysis is the same as the ones used to report in Table 1.
>
> > Multiple epochs training.
>
> We want to clarify that the current statement in the submission about PPO epoch number is actually imprecise: typically PPO does not train for 4 overall epochs, but the 4 epochs refer to the fact that PPO requires 4 epochs RL update for each online minibatch. We will fix the statement in our final version. PPO’s additional computation cost is indicated in the new computation comparison analysis. Since HyPO uses REINFORCE/RLOO for online kl optimization, we observe that taking 1 gradient step per minibatch suffices and multiple epochs per minibatch does not improve performance (however, multiple batches of PPO update seems necessary).
>
>
> > The analysis in this work only focuses on the solution space.
>
> We thank the reviewer for pointing out the additional limitation that our paper only focused on the solution space. We agree that analyzing the more fine-grained dynamics of the algorithms is a very important future direction.

---

> > ### Comment · Reviewer_YCrR · 2024-08-12
> > **Thank you for the response!**
> >
> > I would like to thank the authors for their response. After reading the authors' response and other reviewers' comments, I believe my initial evaluation is appropriate and thus will maintain my rating.

---

### Author Rebuttal · Authors · 2024-08-06

### General responses

We thank all reviewers for their positive and constructive feedback. In the general response we provide some additional empirical results, which will be incorperated in our final version of the paper.

1. **Large-scale experiments:**
In response to Reviewer ewpX’s suggestion on experiments on larger model and additional dataset, we finetune Llama 3 8B instruct model on the Ultrafeedback dataset [1] and evaluate on AlpacaEval [2], MT-Bench [3] and Open LLM Leaderboard, the standard large-scale experiment setup for empirical RLHF papers [4,5]. Following [4], we only finetune the last 4 layers (same for the DPO baseline), and we trained on 8 A100 GPUs. For this experiment, we use RLOO [6] to optimize the online trajectory level KL with $k=2$ (number of repeated online generation). We summarize the results in Table 2 and 3, and we provide the hyperparameter for HyPO with RLOO in Table 4 of the rebuttal supplemental material.

2. **Comparison with PPO and computational analysis:**
In response to Reviewer YCrR’s suggestion we updated our TL;DR result with comparison with PPO: the updated Table is in Table 1 of the supplementary material. Note that as we mentioned in the submission, there is still a gap between HyPO and PPO, and providing a more in-depth theoretical analysis of such gap is an interesting future direction. In addition, we performed a controlled experiment on the computational cost of each method. Following the setup of [4], we fix the computational resource on the same node with 8 A6000 GPUs, and we run each algorithm for a batch of 512 prompts of the TL;DR dataset, averaged over 50 batches. We present the result in Figure 1 of the supplemental material: although HyPO introduces more generation cost, but the time and memory cost of HyPO is still significantly lower than PPO. The hyperparameter of PPO in this experiment is recorded in Table 5 of the supplemental material.

We believe the additional empirical results can further validate our theoretical results, and the large-scale experiment can also have independent empirical contribution to the community. We appreciate the reviewers for their suggestions and we look forward to further feedbacks during the discussion period.

[1] Ultrafeedback: Boosting language models with high-quality feedback. 2023

[2] Length-controlled alpacaeval: A simple way to debias automatic evaluators. 2024

[3] Judging llm-as-a-judge with mt-bench and chatbot arena. 2024

[4]  Rebel: Reinforcement learning via regressing relative rewards. 2024

[5] SimPO: Simple Preference Optimization with a Reference-Free Reward. 2024

[6] Buy 4 REINFORCE samples, get a baseline for free! 2019

---

### Decision · Program_Chairs · 2024-09-25

**Decision:**

Accept (poster)

**Comment:**

This paper studies the performance gaps between online and offline preference alignment methods, and uses the notions of strong and weak dataset coverage to provide explanations. A Hybrid PO algorithm is introduced to use online samples for calculating KL divergence, which gains some performance improvement over original DPO. All reviewers agreed that the story is complete and the contributions are solid, and I concur.